# Multi-scale modeling of urban air pollution: development and application of a Street-in-Grid model (v1.0) by coupling MUNICH (v1.0) and POLAIR3D (v1.8.1)

Youngseob Kim[1], You Wu[2], Christian Seigneur[1], and Yelva Roustan[1]

[1]CEREA, Joint Laboratory École des Ponts ParisTech / EDF R&D, Université Paris-Est, 77455 Champs-sur-Marne, France
[2]EDF R&D China, 100005 Beijing, China

*Correspondence to:* Youngseob Kim (youngseob.kim@enpc.fr)

**Abstract.** A new multi-scale model of urban air pollution is presented. This model combines a chemical-transport model (CTM) that includes a comprehensive treatment of atmospheric chemistry and transport at spatial scales down to 1 km and a street-network model that describes the atmospheric concentrations of pollutants in an urban street network. The street-network model is the Model of Urban Network of Intersecting Canyons and Highways (MUNICH), which consists of two main components: a street-canyon component and a street-intersection component. MUNICH is coupled to the POLAIR3D CTM of the Polyphemus air quality modeling platform to constitute a Street-in-Grid (SinG) model. MUNICH is used to simulate the concentrations of the chemical species in the urban canopy, which is located in the lowest layer of POLAIR3D, and the simulation of pollutant concentrations above roof-tops is performed by POLAIR3D. Interactions between MUNICH and POLAIR3D occur at roof level and depend on a vertical mass transfer coefficient that is a function of atmospheric turbulence. SinG is used to simulate the concentrations of nitrogen oxides ($NO_x$) and ozone ($O_3$) in a Paris suburb. Simulated concentrations are compared to $NO_x$ concentrations measured at two monitoring stations within a street canyon. SinG shows better performance than MUNICH for nitrogen dioxide ($NO_2$) concentrations. However, both SinG and MUNICH underestimate $NO_x$. For the case study considered, the model performance for $NO_x$ concentrations is not sensitive to using a complex chemistry model in MUNICH and the Leighton $NO/NO_2/O_3$ set of reactions is sufficient.

## 1 Introduction

Urban air pollution has been a public health issue for many decades. Historically, the first urban air quality model with spatial and temporal resolution was developed for the Los Angeles basin in California, USA (Reynolds et al., 1973). This three-dimensional (3D) gridded Eulerian model used the atmospheric diffusion (mass-conserving) equation to calculate the change with respect to time of the relevant air pollutant concentrations due to emissions, transport, chemical transformation, and deposition. Because of the urban design of western U.S. cities, there was no need to take buildings into account explicitly.

European cities differ from the Los Angeles basin because of the presence of densely built districts with street-canyon configurations. Consequently, although air quality models such as the one initially used for the Los Angeles basin are commonly used to calculate urban background pollution, different types of air quality models are needed to calculate air pollution at the

street scale. The conceptual approach of the Operational Street Pollution Model (OSPM) has typically been used (Berkowicz, 2000). The air pollutant concentrations are calculated within a street-canyon assuming uniform traffic emissions across the street-canyon, but air pollutant concentrations can be calculated in ventilated and recirculated zones of the street-canyon. Mass transfer between the street and the urban background atmosphere at the top of the street (i.e., roof level) is simulated.

This initial concept has been extended to calculate air pollutant concentrations within a network of streets with the SIRANE model (Soulhac et al., 2011). Although the SIRANE formulation does not distinguish recirculation and ventilation zones and assumes a uniform concentration for each street segment, it provides quite a better treatment of pollutant transport across street intersections. The development of the SIRANE formulation is based on a comprehensive investigation of airflow and mass transfer via wind tunnel experiments and computational fluid dynamics (CFD) simulations. SIRANE has been applied to
various urban districts and has shown satisfactory performance when compared to ambient air pollutant concentrations (e.g. Soulhac et al., 2012). However, the treatment of the urban background above roof level in SIRANE is modeled using a Gaussian model formulation, which prevents the use of a comprehensive atmospheric chemistry. Consequently, it is not appropriate to simulate secondary air pollutants such as ozone ($O_3$) or fine particulate matter ($PM_{2.5}$), which require modeling the formation of secondary pollutants with a comprehensive chemical kinetic mechanism.

Therefore, there is a dire need to combine the advantages of 3D gridded Eulerian models, which can simulate urban background concentrations of all major air pollutants of interest, and those of street-network models, which can simulate the concentrations of air pollutants in complex urban canopy configurations. The multi-scale combination of Eulerian models with near-source models was developed initially for the treatment of plumes from tall stacks in the Los Angeles basin (Seigneur et al., 1983). Many other "Plume-in-Grid" (PinG) models have been developed over the following three decades (see Karam-
chandani et al., 2011, for an overview). Later PinG model development efforts have included PinG models for line sources, area sources, and volume sources using various modeling approaches (e.g., Cariolle et al., 2009; Karamchandani et al., 2009; Huszar et al., 2010; Jacobson et al., 2011; Briant and Seigneur, 2013; Holmes et al., 2014; Kim et al., 2014) in order to treat aircraft emissions, ship emissions, traffic emissions from roadways, and fugitive emissions from industrial sites. However, there is currently no integrated model that dynamically combines an Eulerian model with a street-network model. The objective of this
work is to develop the formulation of such a Street-in-Grid model (SinG), fully consistent with the mass conservation principle, and present its initial application to an actual urban case study. The Eulerian host model selected for this work is POLAIR3D of the Polyphemus air quality modeling platform (Mallet et al., 2007), a 3D chemical-transport model (CTM), which has been widely applied in Europe, North America, South America, Asia, and Africa (e.g., Sartelet et al., 2012). The Model of Urban Network of Intersecting Canyons and Highways (MUNICH), which is used to simulate subgrid concentrations in the urban
canopy represented by the street network, is presented in the next section. Then, the coupling of MUNICH to POLAIR3D is described in Section 3. Finally, some initial applications of MUNICH and the SinG model to a Paris suburb are discussed.

## 2 Description of MUNICH

MUNICH is based conceptually on the SIRANE general formulation (Soulhac et al., 2011). We can distinguish two main components to MUNICH: (1) the street-canyon component, which represents the atmospheric processes in the volume of the urban canopy, and (2) the street-intersection component, which represents the processes in the volume of the intersection. These components are connected to the POLAIR3D model at roof level and are also interconnected. We describe each one of these components in turn.

### 2.1 Street-canyon component

For a street segment, which is defined as a street component bounded by intersections with other streets at each end, the following assumptions are used (Soulhac et al., 2011):

- Air pollutant concentrations are uniform within a street segment.
- The width of the street and the height of the buildings are uniform.
- Emissions of air pollutants and deposition of air pollutants are uniform along the street segment. However, deposition fluxes to different surfaces, including pavement, building walls, and roofs are distinguished using the urban dry deposition model of Cherin et al. (2015).
- The wind direction follows the street segment direction.
- The wind speed is uniform and is related to the wind speed at roof level, the angle between the wind direction at roof level and the street segment direction, and the street segment characteristics (width and height).
- Steady state is assumed for a given time step.

Assuming steady state, the mass flux ($Q$ in $\mu g\,s^{-1}$) balance is applied to calculate the concentration of an air pollutant in a street segment.

$$Q_{\mathrm{s}} + Q_{\mathrm{inflow}} + Q_{\mathrm{chem}} \;=\; Q_{\mathrm{vert}} + Q_{\mathrm{outflow}} + Q_{\mathrm{dep}} \tag{1}$$

where $Q_{\mathrm{s}}$ is the source emission rate, $Q_{\mathrm{inflow}}$ is the inflow rate of the air pollutant entering the street from upwind (typically via an intersection), $Q_{\mathrm{vert}}$ is the vertical flux by turbulent diffusion at roof level (see Section 2.1.1), $Q_{\mathrm{outflow}}$ is the outflow rate of the air pollutant leaving the street in the downwind direction, $Q_{\mathrm{dep}}$ is the pollutant loss rate due to atmospheric deposition, and $Q_{\mathrm{chem}}$ is the air pollutant chemical transformation rate (positive for formation and negative for destruction). The emission term, $Q_{\mathrm{s}}$, is obtained typically from a traffic emission model. The inflow term, $Q_{\mathrm{inflow}}$, is obtained from the street-intersection component (see Section 2.2). The outflow rate, $Q_{\mathrm{outflow}}$ is calculated as follows:

$$Q_{\mathrm{outflow}} \;=\; HW u_{\mathrm{street}} C_{\mathrm{street}} \tag{2}$$

where $H$ is the mean building height in the street segment and $W$ is the mean street width, $u_{\mathrm{street}}$ is the mean horizontal wind velocity in the street segment (see Section 2.1.2), and $C_{\mathrm{street}}$ is the air pollutant concentration in the street segment.

### 2.1.1 Turbulent vertical mass transfer at the top of the street segment

The vertical flux, $Q_\text{vert}$, as formulated in SIRANE does not depend on the building height in the street segment and is, therefore, defined by the external flow condition, based on Salizzoni et al. (2009).

$$Q_\text{vert} = \frac{\sigma_\text{w} W L}{\sqrt{2\pi}} \left( C_\text{street} - C_\text{background} \right) \tag{3}$$

where $C_\text{background}$ is the mean concentration above the street segment, $L$ is the street length, and $\sigma_\text{w}$ is the standard deviation of the vertical wind velocity at roof level, which depends on atmospheric stability. One notes that this approach represents the turbulent mass transfer rate using a mass transfer coefficient with unit of a velocity. Such an approach is routinely used in engineering where mass transfer coefficients are empirically defined and combined with concentration gradients to calculate mass transfer rates. In air quality modeling, this approach is also used to model dry deposition and turbulent mass transfer in the surface layer is typically approximated with a deposition velocity.

A slightly different parametrization was recently proposed by Schulte et al. (2015) who used a turbulent dispersion coefficient defined as follows:

$$K_m = \sigma_\text{w} l \tag{4}$$

where $l$ is a characteristic mixing length within the street-canyon. By assuming that the size of the large turbulent eddies dominating vertical mixing is limited by the smaller size of the street width and height, $l$ is proportional to the smaller of $W$ and $H$ as follows.

$$\frac{1}{l} \sim \left( \frac{1}{W} + \frac{1}{H} \right) \tag{5}$$

Then

$$l = \beta_1 \frac{W H}{W + H} = \beta_1 H \frac{1}{1 + a_r} \tag{6}$$

where $\beta_1$ is a constant and $a_r$ is the aspect ratio (ratio of building height to street width, $H/W$) (Landsberg, 1981).

Then, the vertical flux at roof level is expressed using the turbulent dispersion coefficient as follows:

$$Q_\text{vert} = \beta_2 K_m \frac{W L}{H} \left( C_\text{street} - C_\text{background} \right) \tag{7}$$

By combing Equation 7 with Equations 4 and 6, we obtain

$$Q_\text{vert} = \beta \sigma_\text{w} W L \left( \frac{1}{1 + a_r} \right) \left( C_\text{street} - C_\text{background} \right) \tag{8}$$

where $\beta = \beta_1 \beta_2$.

The constant $\beta$ can be estimated by comparison to Equation 3. Because the vertical flux in Equation 3 is estimated using the unity aspect ratio ($a_r = 1$), we assume that the computed vertical fluxes with Equations 3 and 8 are equal when $a_r = 1$. We obtain $\beta = 0.45$. Figure 1 compares the vertical transfer coefficient estimated with Equations 3 and 8. If $a_r < 1$, i.e., in an area with low buildings, then the transfer coefficient is greater with the formulation of Schulte et al. (2015) than that of SIRANE. On the contrary, if $a_r > 1$, i.e., in a street-canyon configuration, then the vertical transfer is reduced compared to that of SIRANE.

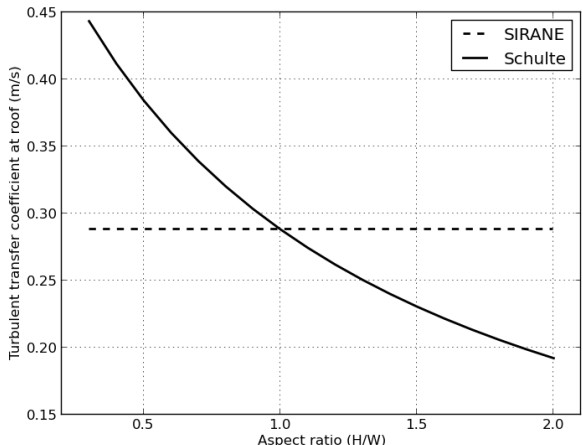

**Figure 1.** Comparison of the turbulent transfer coefficients of the SIRANE formulation (dotted line) and the formulation of Schulte et al. (2015) (solid line).

### 2.1.2 Mean wind velocity within the street-canyon

Here, we use the exponential wind vertical profile proposed by Lemonsu et al. (2004) and used by Cherin et al. (2015) in their modeling of dry deposition within street-canyons. The corresponding formulas were modified here to be specific to the angle between the wind direction and the street-canyon direction (Lemonsu et al., 2004 and Cherin et al., 2015 averaged the wind profile over all possible angles).

  – For narrow canyons, $a_r > 2/3$:

$$u_{\text{street}} = \frac{2}{\pi} u_{\text{H}} cos(\varphi) exp \left( \frac{a_r}{2} \left( \frac{z}{H} - 1 \right) \right) \tag{9}$$

  where $\varphi$ is the angle between the wind direction above roof level and the street direction. $u_{\text{H}}$ is the wind speed at the building height and is a function of the friction velocity.

  – For the so-called intermediate case (i.e., moderate canyons), $1/3 \leq a_r \leq 2/3$:

$$u_{\text{street}} = \left[ 1 + 3 \left( \frac{2}{\pi} - 1 \right) \left( \frac{H}{W} - \frac{1}{3} \right) \right] u_{\text{H}} cos(\varphi) exp \left( \frac{a_r}{2} \left( \frac{z}{H} - 1 \right) \right) \tag{10}$$

  – For a wide configuration, $a_r < 1/3$:

$$u_{\text{street}} = u_{\text{H}} cos(\varphi) exp \left( \frac{a_r}{2} \left( \frac{z}{H} - 1 \right) \right) \tag{11}$$

An average wind speed can be derived from these empirical wind profiles by integrating over the entire street-canyon height $(0 < z < H)$. These empirical wind profiles are exponential functions and are, therefore, qualitatively similar to the profile used in SIRANE (Soulhac et al., 2008) to derive the average wind velocity within the street-canyon. The wind speeds calculated

using these wind profiles and those in SIRANE are compared in Figure 2. This figure illustrates the differences in the mean wind speed obtained for different values of the aspect ratio ranging from 0.1 to 2. The largest differences are obtained when $a_r = 2/3$ and the angle between the wind direction and the street direction is lower. For $\varphi = 0$, the average wind speed of MUNICH is about 2/3 that of SIRANE.

Wind speed observations were not available to compare the results of the two methods. However, due to the relatively low aspect ratio of the street considered in this study ($a_r \sim 1/3$ for Boulevard Alsace-Lorraine), we do not expect to have a strong sensitivity to the choice of the formulation for the average wind speed. This point could become more crucial for streets with higher aspect ratio and should be considered for future applications.

## 2.2    Street-intersection component

The street-intersection component of MUNICH involves the following assumptions, also used in SIRANE (Soulhac et al., 2009):

- – The air pollutant concentration is not uniform across the intersection (as it has sometimes been assumed in earlier work).
- – The advective air flow in the street network is compensated by inflow or outflow at the top (roof level) of the intersection to ensure mass balance.
– The mean air flow follows the wind direction at roof level.
- – The streamlines of the flow from a street to other streets across the intersection cannot cross one another.
- – Fluctuations in wind direction are taken into account when constructing the air flows from one street to others across the intersection.

     Accordingly, the air mass fluxes (and the associated pollutant mass fluxes) are computed for the streets that are connected to 20 the intersection (entering or leaving the intersection) using Equation 1. The air mass fluxes for the streets are corrected by the computed vertical air flux in the intersection at roof level.

     If one considers only the mean air flow, the air flow rates for the streets are determined solely based on the configuration of the streets, their intersection and the wind direction above roof level. However, experiments in a wind tunnel and CFD simulations have shown that fluctuations in wind direction influence importantly the air flow across an intersection (Soulhac 25 et al., 2009). Accordingly, one must take into account these fluctuations to properly account for the transfer of air (and pollutant) mass across the intersection. Then, the computation of the air fluxes depends not only on the mean wind direction, but also on the wind fluctuation. The wind direction is assumed to follow a Gaussian distribution centered on its mean value.

## 2.3    Chemical reactions

In MUNICH, the CB05 chemical kinetic mechanism (Yarwood et al., 2005) is implemented to ensure consistency with PO- 30 LAIR3D in the SinG configuration. CB05 consists of 53 species including volatile organic compounds (VOC) and inorganic species and 155 chemical reactions including 23 photolytic reactions. However, nitric oxide (NO) emissions in the urban canopy are likely to scavenge $O_3$ and other oxidants, thereby suppressing VOC chemistry. Accordingly, a simple three-reaction mechanism involving solely NO, nitrogen dioxide ($NO_2$) and $O_3$, known as the Leighton photostationary state (Leighton, 1961),

was also implemented. However the Leighton photostationary state may not hold even in urban environment when VOC emissions are high (Trebs et al., 2012; Matsumoto et al., 2006). These two mechanisms are compared below in terms of model performance and computational costs.

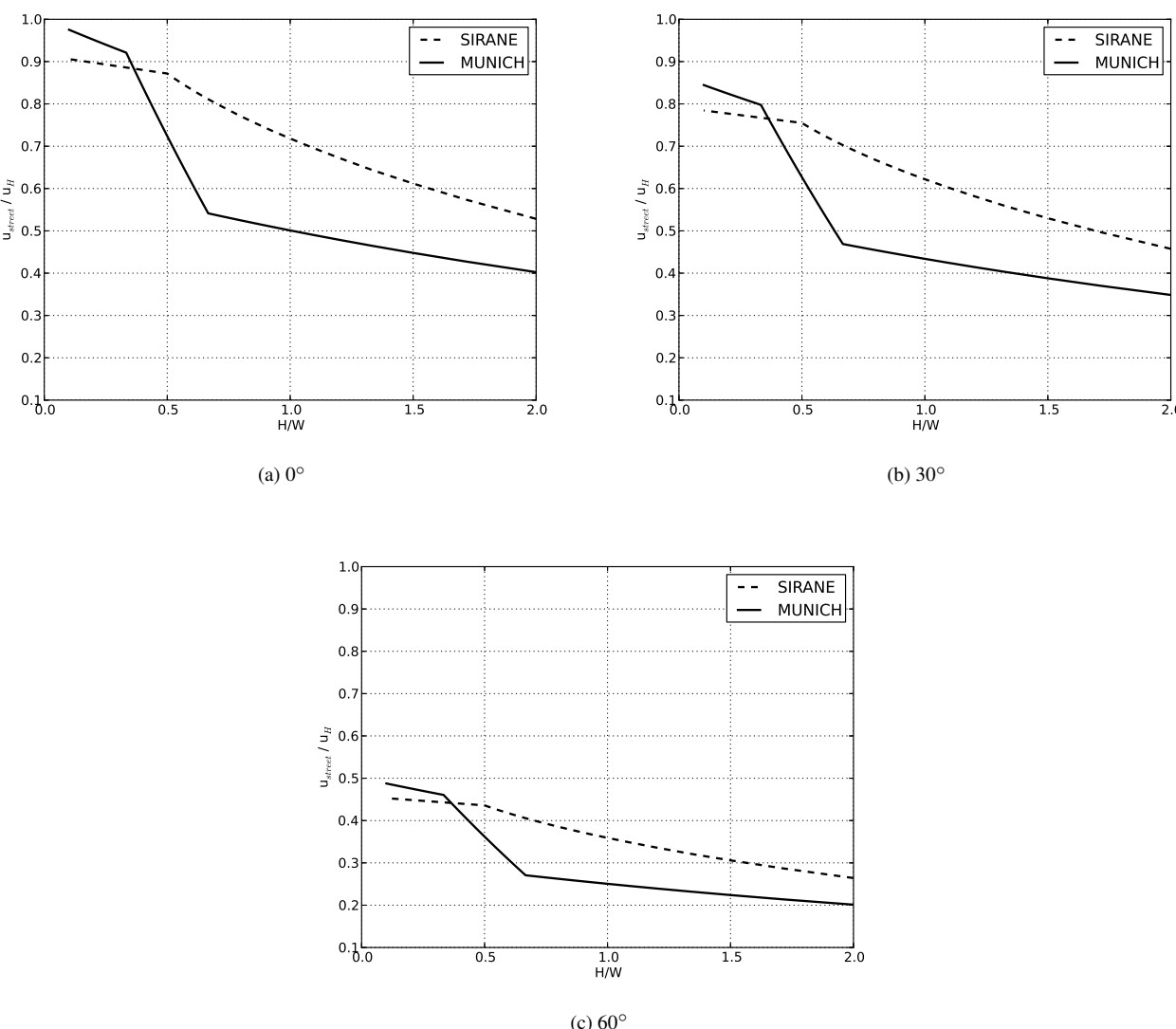

(a) 0°

(b) 30°

(c) 60°

**Figure 2.** Comparison of the mean horizontal wind velocity (normalized with respect to the wind speed at roof level) within the street-canyon calculated with the profiles of SIRANE (Soulhac et al., 2008) (dotted lines) and MUNICH (Lemonsu et al., 2004) (solid lines) as a function of the street aspect ratio for three different angles between the wind direction and the street direction (a) 0°, (b) 30°, (c) 60°

## 2.4 Dry and wet deposition

Dry deposition is computed using the approach developed for an urban canopy (Cherin et al., 2015). Surfaces available for dry deposition include pavement (street and sidewalks), building walls, and building roofs. The dry deposition fluxes (in $\mu g \, m^{-2} \, s^{-1}$) are calculated by multiplying the pollutant concentrations (in $\mu g \, m^{-3}$) and the pollutant deposition velocities (in $m \, s^{-1}$). The estimation of the deposition velocities depends on the atmospheric conditions and the surface properties, which differ among the surface types. For the building roofs, the background concentrations over the urban canopy are used, whereas the concentrations within the street network are used for the pavement and building walls.

Wet deposition consists of the scavenging by precipitation and deposition to pavement and building roofs. Wet deposition to the building roofs is estimated by the precipitation intensity and the background concentrations over the urban canopy. The scavenging and deposition to the pavement is computed for the entire atmospheric column and includes both the background concentrations above roof tops and the concentrations within the urban canopy:

$$F_{\text{street}} \;=\; \Lambda \left( C_{\text{street}} H + C_{\text{background}} (z_c - H) \right) \tag{12}$$

where $F_{\text{street}}$ is the wet deposition flux to the pavement ($\mu g \, m^{-2} \, s^{-1}$), $\Lambda$ is the scavenging coefficient ($s^{-1}$), and $z_c$ is the cloud base height (m). The in-cloud wet scavenging is supposed to have a weak impact for the species considered here.

## 2.5 Summary of MUNICH characteristics

The concept of the street-network model MUNICH is close to the one used in SIRANE to represent concentration at the street level. We have introduced several parametrizations for the vertical turbulent flux and the average wind speed. It is however not possible to definitively advocate a specific choice for these parametrizations with the set of observations available within the framework of the TrafiPollu project (http://www.agence-nationale-recherche.fr/?Project=ANR-12-VBDU-0002). MUNICH is then kept modular, the model can rely on the different parametrizations following user choices. MUNICH is designed as a stand-alone street-network model and does not aim to represent concentrations over the urban canopy. Beyond its modularity the main strength of MUNICH over SIRANE relies on the possibility to represent a complex chemistry in the street. It also allows the interactive connection with an Eulerian chemistry transport model.

## 3 Coupling of MUNICH with POLAIR3D: Street-in-Grid model

We describe here a new model, "Street-in-Grid" (SinG), which combines the MUNICH street-network model and the PO-LAIR3D CTM. SinG is conceived to conduct a multi-scale simulation, which estimates both grid-averaged concentrations at the urban scale and concentrations within each street segment. This combined model provides the following advantages.

– It allows one to estimate the influence of the background concentrations on the concentrations within the street network and vice-versa.

– There is no double counting of emissions, originating within the urban canopy: these emissions are input data to MUNICH and, therefore, they are removed from the grid-averaged emission inventory of POLAIR3D.

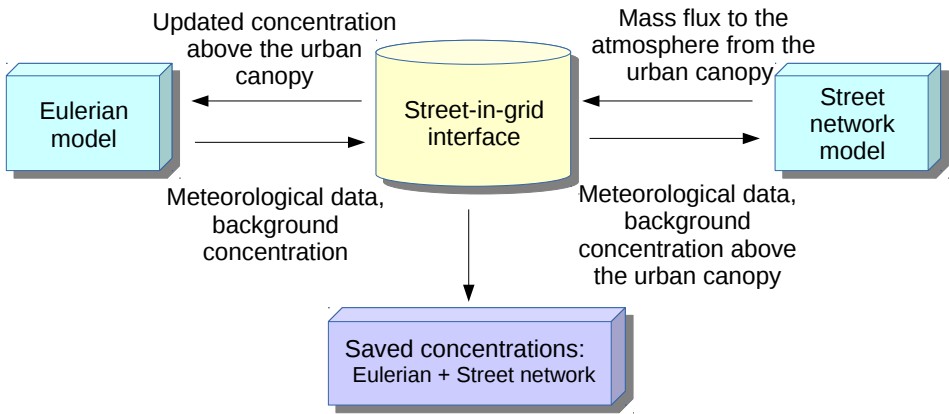

**Figure 3.** Schematic diagram of the Street-in-Grid model.

– There is consistency between the treatment of physical and chemical processes at different scales. Transport and dispersion of pollutants at the urban and street-network scales are calculated from the same meteorological data. Similarly, the same chemical mechanism and the same formulations for dry and wet atmospheric deposition are used at those different scales. There is, however, the option to use a reduced form of the chemical mechanism within the street network, following Karamchandani et al. (1998).

Figure 3 shows schematically the concept of the SinG model. As MUNICH is located within the lowest POLAIR3D layer, meteorological variables in that layer, such as wind speed and direction, are transferred to MUNICH via the SinG interface. Air pollutant concentrations in the POLAIR3D lowest layer are also transferred since they are used as the background concentrations for the street network. Then, MUNICH computes the mass fluxes between the urban canopy (i.e., the street network) and the urban atmosphere above roof level and the SinG interface transfers them to POLAIR3D to compute new air pollutant concentrations in the grid cells above the urban canopy. The interfacing between MUNICH and POLAIR3D is conducted at fixed time steps, which were set at 10 min in the following application, the integration time step of the Eulerian model.

## 4 Application of MUNICH to a street network in a Paris suburb

### 4.1 Simulation domain and setup

MUNICH was applied to simulate the concentrations of pollutants in a Paris suburb (Le Perreux-sur-Marne, 13 km east of Paris). Figure 4 displays the location of the modeling domain. The street-network within the simulation domain consists of

577 street segments and is displayed in Figure 5. Simulations for gas-phase species including $NO_x$, CO, VOC emissions were conducted during the period from March 24 to June 14, 2014. Here, we use the parametrization proposed by Schulte et al. (2015) for the vertical flux at roof and the exponential wind vertical profile proposed by Lemonsu et al. (2004) for the mean wind speed within the street-canyon.

## 4.2 Traffic emissions

The traffic emission inventory used for the simulation domain was built for the TrafiPollu project. This emission inventory rely on the use of the dynamic traffic model Symuvia (Leclercq et al., 2007) and the COPERT 4 emission factors (http: //emisia.com/products/copert-4/versions). The dynamic traffic model Symuvia calculates the vehicle trajectories, the number of vehicles and the averaged speed on a given time period for each street segment of the simulated street network. Dynamic traffic models represent vehicle flow at smaller spatial and temporal scale than static traffic models and potentially allow an explicit representation of traffic congestion. A discussion on the differences between dynamic and static traffic models in link with water and air quality studies can be found in Shorshani et al. (2015). However for the current work the Symuvia outputs were averaged and combined with COPERT 4 emission factors to generate hourly emission rates for each street segment. The emission rates depend on the averaged vehicle speed and composition of the vehicle fleet. This latter was determined through

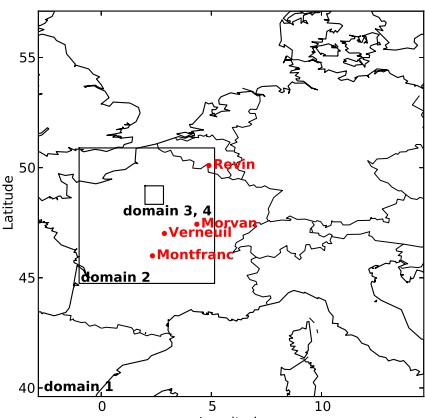 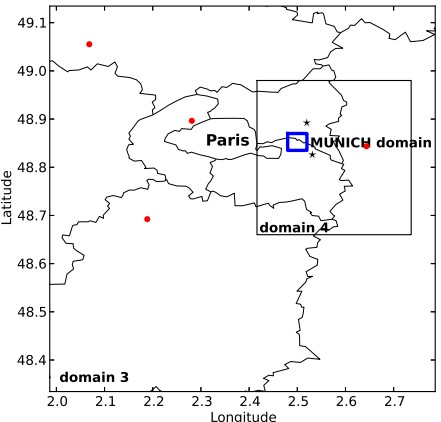

**Figure 4.** Four simulation domains are simulated from the continental scale to the urban scale. In the left panel, the largest domain 1 covers western Europe. Domain 2 covers northern/central France. the red circles show the locations of the background air monitoring stations. In the right panel, domains 3 and 4 cover the Île-de-France region and the eastern Paris suburbs. the blue box corresponds to the modeling area in suburban Paris for the MUNICH simulations. The black stars and red circles show the locations of the urban background air monitoring stations. Measured data at the stations with the black stars are used for background concentrations in the MUNICH simulations. SinG is only used for domain 4.

video monitoring (André et al., 2017). It is however important to notice that the vehicle fleet composition appears to be a sensitive input data (Carteret et al., 2014; Chen et al., 2017).

Two typical days (March 25 for weekday and March 30 for weekend) were chosen to be simulated with the traffic model and used to represent the traffic emission over the whole period. The traffic model estimates the vehicle flow for each traffic direction of a two-way street. The traffic emissions of a two-way street were then merged to obtain one emission rate for each simulated street segment, the basic input data needed by MUNICH.

Surface areas of intersections are not taken explicitly into account in MUNICH and streets are connected at the center of the intersection, i.e., an intersection is represented by a point using a latitude/longitude coordinate set. The geometry of the intersection can influence the mass exchange (Salem et al., 2015). In particular, when intersections are large, vertical mixing with the overlying atmosphere becomes more important. As this phenomenon is not taken into account in the current version of the model it leads to underestimate the exchanges through such open space in the street network. There is a need here to extend the modeling framework to better represent this type of urban space.

Figure 5 shows the $NO_x$ traffic emissions which were estimated for the 577 street segments of the simulation domain in the Paris suburb. In the left panel, $NO_x$ emission rates during nighttime are presented. Very low emission rates are estimated for all the streets even though those on the A86 highway are slightly higher. In the right panel, $NO_x$ emission rates during morning rush-hour increase more than $1\,400\,\mu g\,m^{-1}\,s^{-1}$. Since the traffic model is calibrated with flow observation and the vehicle fleet composition determined through video monitoring, the remaining uncertainties in the emission data lie in the use of only two typical days to represent the whole period and in COPERT 4 emission factors.

## 4.3 Geographic data

Traffic lane widths and building heights were obtained from the BD TOPO database (http://professionnels.ign.fr/bdtopo). Total street width includes the lane width, the sidewalk width or the highway shoulder width (the A86 highway passes through the modeling domain). For minor surface roads, a width of 3 m was used for sidewalks by default, which corresponds to 2 sidewalks (the minimum sidewalk width in France is 1.4 m). For the A86 highway, 20 m were added to the lane width including 2 shoulders (4 m), a median strip (1.5 m), and 2 urban-train lanes (4 m). Street widths and building heights of the 15 major streets were explicitly estimated. For the other streets, average street width (7.5 m) and building height (6.9 m) estimated for the modeling domain were used.

## 4.4 Meteorological data

Meteorological data, including wind direction/speed, planetary boundary layer (PBL) height, and friction velocity, were obtained from a Weather Research and Forecasting (WRF) model version 3.6.1 (Skamarock et al., 2008) simulation conducted with a horizontal resolution of $1.5 \times 1.5\,km^2$ (Thouron et al., 2017). The simulated meteorological data were compared to the measurements at three urban-background meteorological stations near the simulation domain. The root-mean square error (RMSE), the fractional bias (FB), and the correlation coefficient (R) are the statistical indicators used in Thouron et al. (2017) to evaluate the meteorological fields. The WRF simulation slightly overestimates the temperature (RMSE: $0.2 \sim 1.1\,°C$, FB:

$0.02 \sim 0.07$ and R: 0.9) and overestimate the wind speed (RMSE: $0.8 \sim 1.1\,\mathrm{m\,s^{-1}}$, FB: $0.2 \sim 0.3$ and R: $0.6 \sim 0.7$). The modeled wind direction is biased by an angular differences of about $15°$. An important error in the precipitation modeling is obtained (RMSE: $0.04\,\mathrm{mm\,h^{-1}}$, FB: -0.6, R: 0.1) but this model error has not a strong impact on the concentration of the poorly soluble species simulated.

## 4.5 Background concentrations

Background concentrations of NO, $NO_2$, and $O_3$ were obtained from two urban background air monitoring stations near the modeling area (5 to 7 km from the area, see Figure 4). Averaged values of the hourly measured concentrations at the two stations were used to compute the vertical mass transfer at the top of the street network in Equations 3 and 8. These stations are operated by AIRPARIF, the air quality agency of the Paris region (http://www.airparif.asso.fr/).

## 4.6 Results

Figure 6 shows that simulated concentrations of $NO_x$ are high in the streets where the emission rates are high. The concentrations of $NO_x$ during nighttime on March 25 reach $160\,\mathrm{\mu g\,m^{-3}}$ over the major streets. During the morning rush-hour

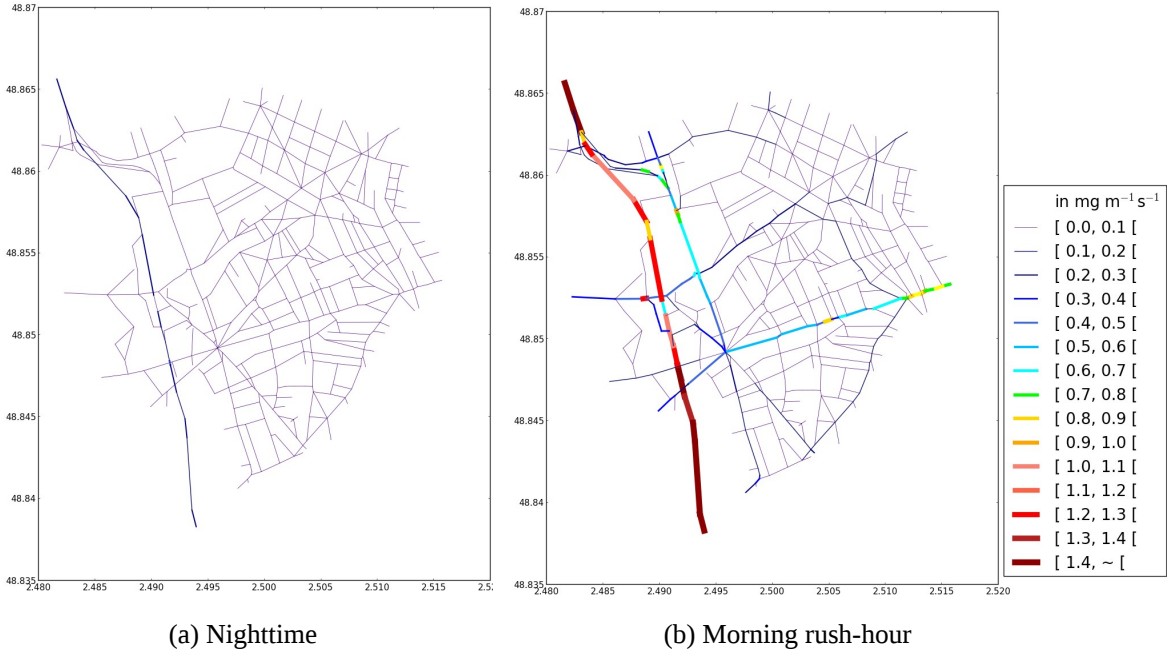

(a) Nighttime        (b) Morning rush-hour

**Figure 5.** $NO_x$ emission rates ($\mathrm{\mu g\,m^{-1}\,s^{-1}}$) used in MUNICH simulations for a week day (a) during nighttime at 1 AM (UTC) (b) in the morning rush-hour at 7 AM (UTC) on March 25, 2014.

on the same day, the concentrations of $NO_x$ increase to $600\,\mu g\,m^{-3}$. The modeled high concentrations during the rush-hour are due not only to high emission rates but also to stable meteorological conditions with low PBL height $(520\,m)$ and wind speed $(2.5\,m\,s^{-1})$. One notes that there is a clear difference between the spatial patterns of the emission maps (Figure 5) and concentration maps (Figure 6). Streets with no or little $NO_x$ emissions display non-negligible $NO_x$ concentrations, thereby highlighting the importance of advective and turbulent transport in the street network.

Figure 7 compares the modeled 24-h averaged concentrations of $NO_2$ with the concentrations measured at the air monitoring stations operated by AIRPARIF during the TrafiPollu project on the two sidewalks of Boulevard Alsace-Lorraine for the period from April 6 to June 15. Mean diurnal variations of $NO_2$ concentrations over this period are presented in Figure 8. Statistical indicators defined in Appendix A for the comparison of hourly concentrations are provided in Table 1. The $NO_2$ modeled concentrations using MUNICH generally underestimate the observations with a mean negative bias of 32%. Simulated morning and evening peaks are delayed compared to the observation. The morning peak of emissions data for the street segment of Boulevard Alsace-Lorraine corresponds in time to the peak of observed concentrations. It is also important to note that in average over the street network the morning peak of emissions data occurs one hour later than in Boulevard Alsace-Lorraine.

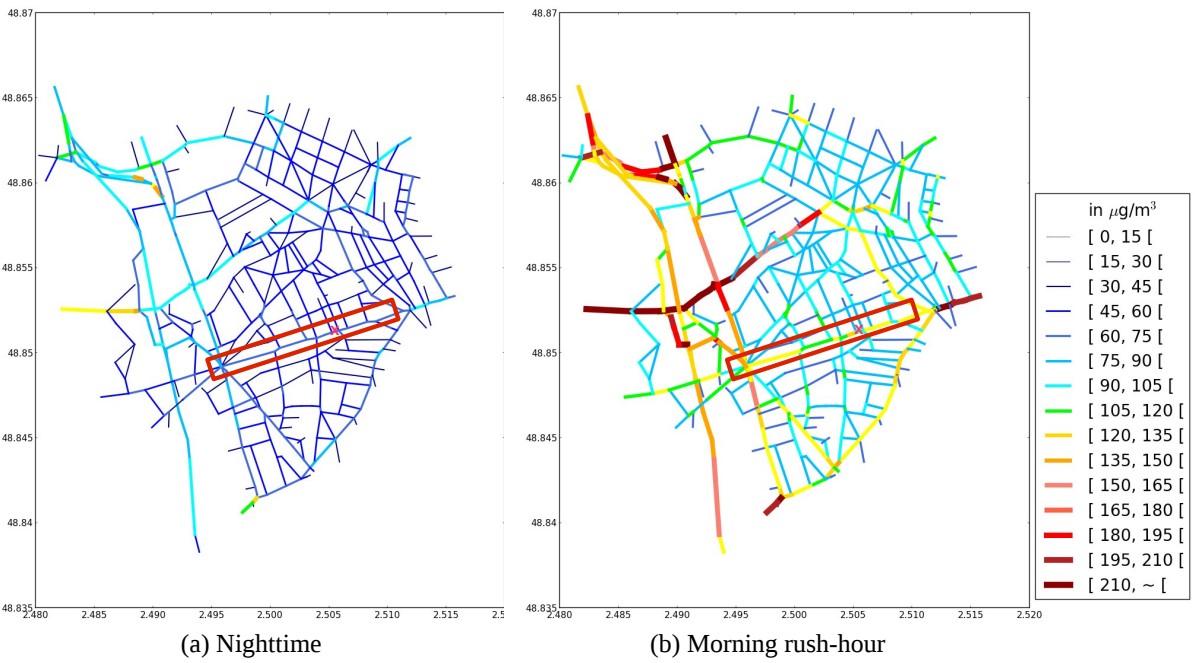

(a) Nighttime

(b) Morning rush-hour

**Figure 6.** Simulated $NO_x$ concentrations using MUNICH (a) during nighttime at 1 AM (UTC) (b) in the morning rush-hour at 7 AM (UTC) on March 25, 2014. The red rectangular box encompasses Boulevard Alsace-Lorraine and the cross mark corresponds to the location of the air monitoring stations on the sidewalks.

It means that the delay in simulated concentrations is introduced by a transport process (advection in the street network or turbulent exchange with the background atmosphere).

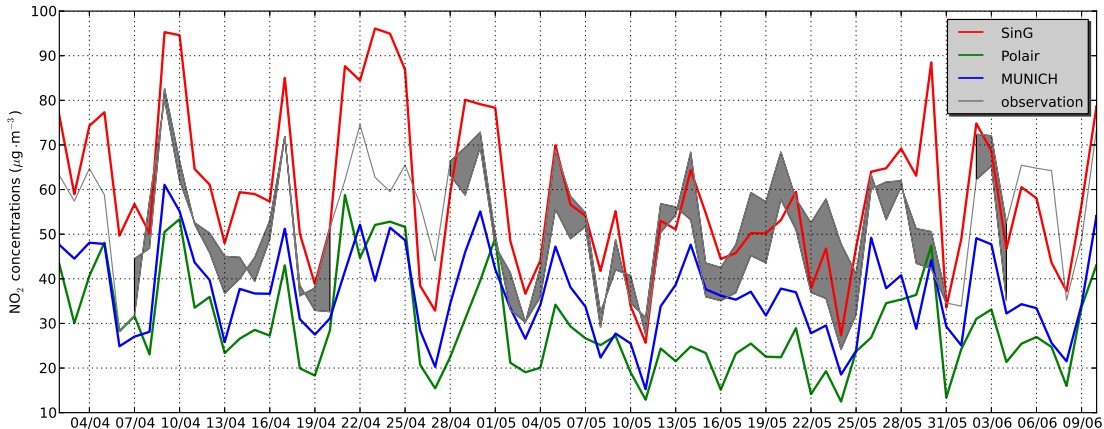

**Figure 7.** Temporal evolution of $NO_2$ daily-averaged concentrations modeled with MUNICH (blue line), POLAIR3D (green line) and the SinG model (red line). They are compared to the measured concentrations (black shaded regions) at the stations nearby traffic on each sidewalks of the Boulevard Alsace-Lorraine. If the measurement is available at only one station, black line is used instead.

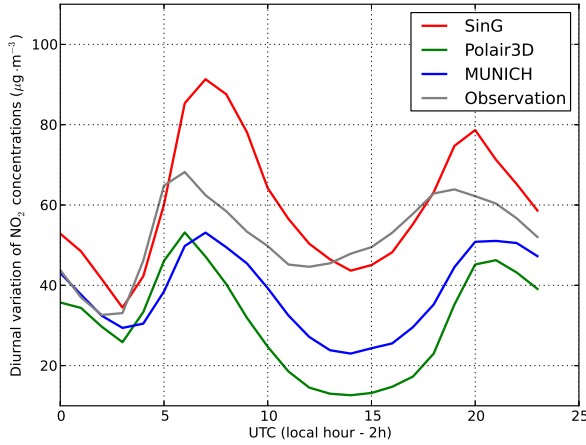

**Figure 8.** Diurnal variation of $NO_2$ concentrations modeled with MUNICH (blue line), POLAIR3D (green line) and the SinG model (red line). They are compared to the measured concentrations (black line) at the stations nearby traffic on each sidewalks of Boulevard Alsace-Lorraine.

**Table 1.** Statistical indicators of the comparison of simulated hourly concentrations to the $NO_2$ and $NO_x$ concentrations measured at the air monitoring stations operated on the sidewalks of Boulevard Alsace-Lorraine.

| | NO$_2$ | | | | | NO$_x$ | | | | |
|---|---|---|---|---|---|---|---|---|---|---|
| | MUNICH | MUNICH-s ** | SinG | SinG-s *** | POLAIR3D | MUNICH | MUNICH-s ** | SinG | SinG-s *** | POLAIR3D |
| Observation ($\mu$g m$^{-3}$) | | | 52.6 | | | | | 148.5 | | |
| Simulation ($\mu$g m$^{-3}$) | 38.1 | 42.2 | 60.2 | 59.7 | 30.8 | 50.3 | 60.8 | 76.8 | 103.7 | 37.4 |
| FB* | -0.32 | -0.22 | 0.13 | 0.13 | -0.52 | -0.99 | -0.84 | -0.64 | -0.36 | -1.19 |
| $\sqrt{NMSE}$* | 0.47 | 0.40 | 0.40 | 0.22 | 0.71 | 1.22 | 1.04 | 0.86 | 0.43 | 1.68 |
| MFE* | 0.42 | 0.35 | 0.31 | 0.19 | 0.67 | 0.99 | 0.87 | 0.64 | 0.39 | 1.15 |
| VG* | 1.35 | 1.23 | 1.17 | 1.06 | 2.30 | 5.24 | 3.58 | 1.96 | 1.25 | 11.89 |
| MG* | 0.69 | 0.78 | 1.12 | 1.14 | 0.51 | 0.32 | 0.38 | 0.52 | 0.69 | 0.24 |
| FAC2* | 0.77 | 0.87 | 0.90 | 1.00 | 0.53 | 0.22 | 0.32 | 0.53 | 0.86 | 0.15 |
| R* | 0.67 | 0.68 | 0.64 | 0.70 | 0.51 | 0.64 | 0.61 | 0.64 | 0.76 | 0.54 |

*: FB (Fractional bias), NMSE (Normal mean square error), MFE (Mean fractional error), VG (Geometrical mean squared variance), MG (Mean geometrical bias), FAC2 (Fraction in a factor of 2), R (Correlation coefficient) (Chang and Hanna, 2004; Yu et al., 2006).

**: For the simulation "MUNICH-s" a 25% reduction of the turbulent transfer coefficient, a one-third increase of $NO_x$ emissions from traffic and a reduction from 20% to 9% of the $NO_2/NO_x$ emissions ratio (in mass of $NO_2$ equivalent) are applied.

***: For the simulation "SinG-s" a 25% reduction of the turbulent transfer coefficient, a 33% reduction of the $O_3$ boundary conditions, a one-third increase of $NO_x$ emissions from traffic and a reduction from 20% to 9% of the $NO_2/NO_x$ emissions ratio (in mass of $NO_2$ equivalent) are applied.

In addition to $NO_2$ concentrations, $NO_x$ concentrations ($NO_2$ equivalent) were measured at the monitoring stations at Boulevard Alsace-Lorraine. The comparison of the measured and simulated concentrations with MUNICH shows a large underestimation in the $NO_x$ concentrations (measurement: $148.5\,\mu$g m$^{-3}$ and simulation with MUNICH: $50.3\,\mu$g m$^{-3}$). Worse model performance for $NO_x$ than for $NO_2$ has also been reported in earlier studies (e.g. Ketzel et al., 2012; Soulhac et al., 2012), which suggests that $NO_2$ model performance may actually benefit from some error compensation. Here for example, the underestimation of $NO_x$ concentrations is partially compensated by an overestimation of the $NO_2/NO_x$ fraction.

It is not obvious to attribute these discrepancies in $NO_2$ and $NO_x$ simulations to uncertainties in the model formulation or the input data (background concentrations, meteorological data and emission data). Nevertheless the sensitivity to the choice of the background concentration is important. For the reference simulation the background concentrations are estimated using the mean of concentrations measured at two urban background stations (see Figure 4). Figure 9 shows similar temporal evolution in the measured $NO_2$ and $NO_x$ daily concentrations between the two stations. However large discrepancies in their

peak values are observed (up to a maximum difference of 300% in the hourly concentrations). It implies that the measured background concentrations certainly do not always correspond to the concentration above a given street. Two additional simulations were conducted to assess the relative contributions from the uncertainties in the background concentrations derived from measurements. For $NO_2$, $NO_x$ and $O_3$ the standard deviations over the simulated period of the differences between the measured concentrations at the two monitoring stations are calculated ($\sigma_{NO}$: $8.1\,\mu g\,m^{-3}$, $\sigma_{NO_2}$: $6.5\,\mu g\,m^{-3}$ and $\sigma_{O_3}$: $5.1\,\mu g\,m^{-3}$). The first simulation was run with $O_3$ concentrations increased by $\sigma_{O_3}$ and NO and $NO_2$ concentrations lowered by $\sigma_{NO}$ and $\sigma_{NO_2}$ respectively. In the second simulation reduced $O_3$ concentration and increased NO and $NO_2$ concentrations are used. Differences between the averaged $NO_2$ concentrations for these simulations and the reference simulation are up to 30%. This result points out the difficulty of identifying measurements that are truly representative of the "urban background" as needed in the street-network model. As shown in the following the urban background concentrations can be estimated based on the concentrations simulated with an Eulerian model. This does not ensure a better representativity of the simulated background concentrations. However a dynamic coupling at least ensures a consistent treatment of the mass conservation. Furthermore it allows scenario analysis in a prospective framework with a consistent evolution of background and local concentrations.

Beyond the urban background concentrations the main remaining uncertainties are related to the evaluation of the vertical transfer at roof top and to the traffic emissions data. A sensitivity test was conducted for further investigation on the $NO_x$ underestimation and the $NO_2/NO_x$ ratio overestimation with a different configuration settings and input data set (MUNICH-s in Table 1). The aim is to propose a first illustration of the uncertainties. A potential underestimation of the $NO_x$ emissions from traffic and an overestimation of the vertical flux by turbulent diffusion at roof level were considered to explain the deficit of $NO_x$ concentrations within the street. The $NO_2/NO_x$ emission ratio is also considered to explain the too high concentration ratio :

– The turbulent transfer coefficient is decreased by 25%.

– A one-third increase of $NO_x$ emissions from traffic is applied in the street network.

– A reduction from 20% to 9% of the $NO_2/NO_x$ ratio (in mass of $NO_2$ equivalent) in the emissions from traffic.

The magnitude of the turbulent transfer coefficient reduction is somewhat arbitrary. It is however chosen consistent with the difference between the two parametrizations considered for the vertical turbulent transfer (Figure 1) for the aspect ratio of Boulevard Alsace-Lorraine. It could account for the uncertainties in the meteorological fields since the standard deviation of the vertical wind velocity ($\sigma_w$) depends on the friction velocity, the Monin-Obukhov length and PBLH that also contribute to the global uncertainty. This reduction can also be seen as a stopgap to deal with the discrepancies due to the assumption of uniform concentration within each street segment. For $NO_x$, mainly emitted near from the street ground, this latter assumption certainly leads to overestimate the concentration at the roof level since the vertical profile of concentrations is rather supposed to be exponentially decreasing with height (Vardoulakis et al., 2003, due to chemistry this may be not the case for NO or $NO_2$ taken separately). This last assumption lead to overestimate the vertical turbulent flux computation for $NO_x$ as a whole. It is interesting to note that beyond the limitation of the $NO_x$ flux toward the background, the decrease of the turbulent transfer

coefficient also improve the $NO_2/NO_x$ concentration ratio. It limits the $O_3$ flux from the background and the mix with an air mass with a larger $NO_2/NO_x$ concentration ratio (observed ratio $\sim$1/3 in the street against $\sim$4/5 in the background).

The increase of emissions is consistent with the uncertainties concerning $NO_x$ emissions derived from COPERT 4 (Kouridis et al., 2010). The value chosen initially for the $NO_2/NO_x$ ratio in the emissions from traffic was determined from roadside concentration observed in Île-de-France (AIRPARIF, 2015). However this value may be not really representative of the tailpipe ratio (Kimbrough et al., 2017). The 9% ratio (value applied for others emissions sectors, Sartelet et al., 2007) appears in the range of possible values reported by Carslaw and Rhys-Tyler (2013).

These modifications of the reference simulation setup improve the $NO_2$ and $NO_x$ concentrations but the $NO_x$ concentrations remain largely underestimated. The sensitivity of the model results to the turbulent transfer coefficient imply that the choice between the Salizzoni et al. (2009) formulation and the one proposed in Schulte et al. (2015) can have an impact for streets with an aspect ratio far from 1. More comprehensive studies need to be conducted for these conditions of aspect ratio (e.g., in Paris center).

## 5 Application of SinG to a street network in a Paris suburb

### 5.1 Simulation domains and input data

SinG is used to estimate the pollutant concentrations in both the 3D gridded domain and the street network. Four simulation domains are used from the continental scale to the urban scale (see Figure 4). Domain 1 covers western Europe with a horizontal

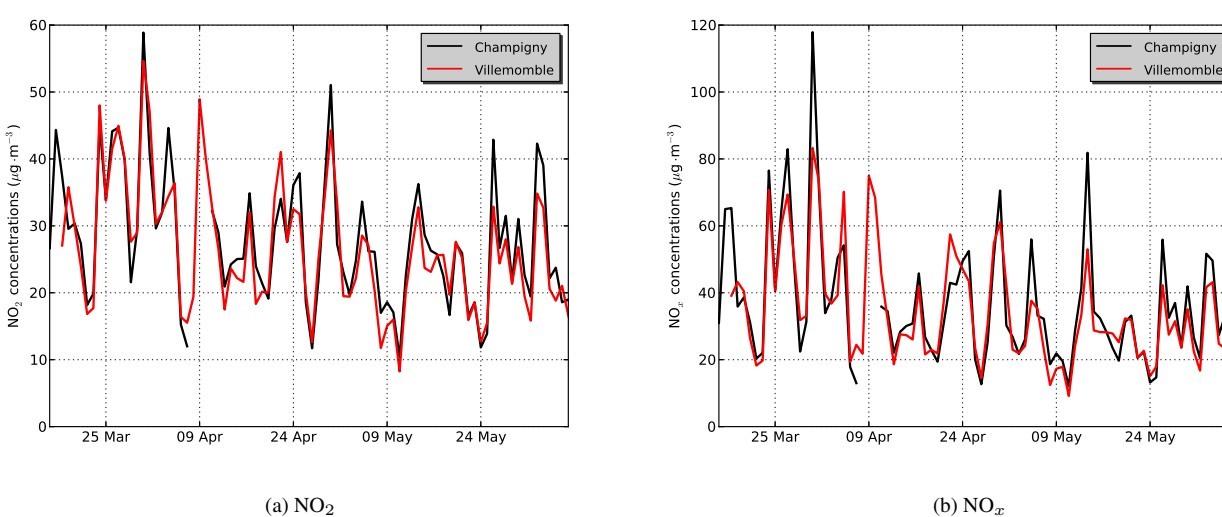

(a) $NO_2$           (b) $NO_x$

**Figure 9.** Comparison of the daily-averaged measurements at the two air monitoring stations for (a) $NO_2$ and (b) $NO_x$. The first station is located at 5 km from the modeling area (Champigny) and the second station is located at 7 km from the modeling area (Villemomble).

resolution of 0.5°. Domains 2 and 3 cover northern/central France (0.15° resolution) and the Île-de-France region (0.04° resolution), respectively. The urban-scale domain 4 covers the eastern Paris suburbs (0.01° resolution) including the area where the street network is located. The horizontal resolution of domain 4 corresponds to about 1 km. The street network neighborhood is covered by 12 grid cells of domain 4 and corresponds to about 1% of the domain 4 area. The vertical resolution consists of

10 levels up to 6 km with the lowest level at 15 m.

For POLAIR3D, boundary conditions for the outer domain 1 were obtained from data simulated by the MOZART 4 global CTM (Emmons et al., 2010). Meteorological data were obtained from WRF simulations for all domains (Thouron et al., 2017). Anthropogenic emissions were calculated using the European Monitoring and Evaluation Programme (EMEP) inventory for domains 1 and 2 (EMEP/CEIP 2014 present state of emissions as used in EMEP models) and the AIRPARIF inventory for

domains 3 and 4. Biogenic emissions were calculated with MEGAN v2.04 (Guenther et al., 2006). For MUNICH, which here is the urban canopy model embedded into POLAIR3D, the input data presented in Section 4 were used, except for boundary conditions over roof top, which were obtained from the lowest layer of POLAIR3D in the SinG simulation.

## 5.2   Evaluation of the simulated background concentrations

Two simulations were performed over domain 4 from March 24 to June 14, 2014. POLAIR3D is used in the first simulation

whereas SinG is used in the second simulation to estimate the influence of the subgrid-scale treatment of the urban canopy on the pollutant concentrations. The background concentrations in the simulation with SinG are modeled by the Eulerian

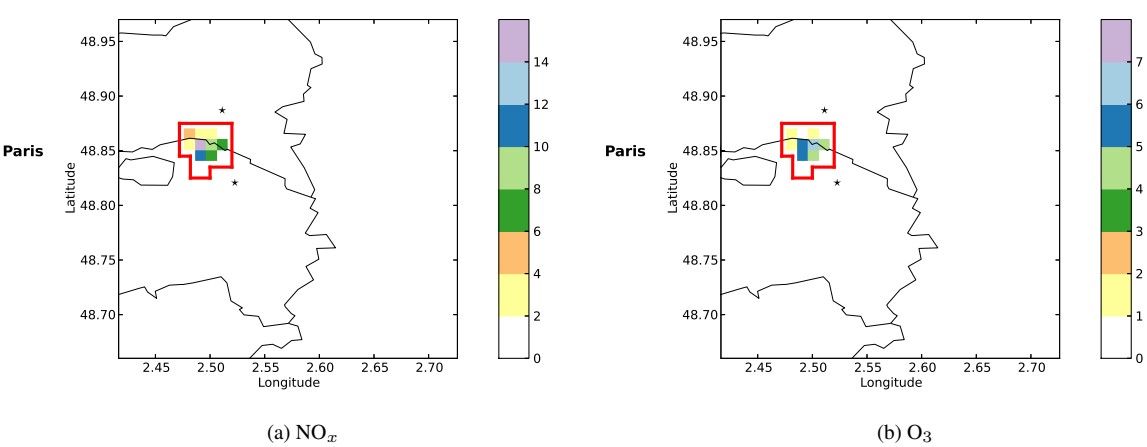

(a) NO$_x$

(b) O$_3$

**Figure 10.** Differences between SinG and POLAIR3D in the surface concentrations (in % for the means over the whole simulation period) of (a) NO$_x$ and (b) O$_3$. The red-boundary enclosed area corresponds to the grid cells where the street network is located. Grid cell concentrations were calculated by combining the street-network and above-roof-top concentrations weighted by the corresponding volumes. The stars show the locations of the urban background air monitoring stations.

model and updated every 10 min during the simulation to provide the needed upper boundary condition to the urban canopy module. The simulated background concentrations of $O_3$ and $NO_x$ by POLAIR3D and SinG are compared to the measured concentrations at the urban background air monitoring stations (Champigny and Villemomble). Because these stations are relatively far from the considered street network, the difference between the two models are limited (see Figure 10). We

obtained satisfactory results in the $NO_x$ and $NO_2$ concentrations but the $O_3$ concentrations are overestimated ($\sim 25\,\mu g\,m^{-3}$ - $\sim 45\%$) at both stations (see Appendix B). The overestimation of ozone concentrations is partly related to an overestimation of the boundary conditions. A comparison of simulated $O_3$ concentrations within domain 3 with the observations at six urban sites of the AIRPARIF network shows an overestimation of around $\sim 25$ - $30\,\mu g\,m^{-3}$ ($\sim 33\%$) (see Appendix B).

Figure 10 presents the differences between the two simulations in the mean concentrations over the whole simulated period

of $NO_x$ and $O_3$. Differences between POLAIR3D and SinG in the $NO_x$ concentrations are at most 15%. These differences are due to different dispersion of $NO_x$ emitted within the urban canopy in SinG and POLAIR3D. Since the wind speed is lower within the urban canopy than above it, advection is slower on average in SinG than in POLAIR3D for the grid cell, that are treated with the urban canopy module. An increase in the $O_3$ concentrations occurs with SinG compared to POLAIR3D (5%). It is due to less $O_3$ titration in SinG than in POLAIR3D. In SinG, vertical dispersion of $NO_x$ is constrained by the urban canopy.

Therefore, $O_3$ titration is less in SinG in comparison to POLAIR3D due to lower NO concentrations above the urban canopy.

## 5.3  Evaluation of the simulated concentrations within the street

For the street segment where measurements are available, the temporal evolution of the modeled $NO_2$ concentrations using SinG is compared to those of MUNICH in Figure 7 and Table 1. Statistical scores in Table 1 show better performance for SinG than MUNICH. The simulated background concentrations affect the concentrations in the street-canyon and lead to better

performance with the current configuration. A similar conclusion was reached by Briant and Seigneur (2013) who compared a PinG model to a gaussian model for simulating $NO_2$ concentrations near roadways. Simulating the background can lead to better performance than using background concentrations from monitoring stations that may not be representative for the considered neighborhood. As expected, the concentrations simulated with the POLAIR3D CTM underestimates the street-canyon $NO_2$ and $NO_x$ concentrations.

The comparison of the measured and simulated concentrations with SinG still show a large underestimation in the $NO_x$ concentrations (measurement: $148.5\,\mu g\,m^{-3}$ and simulation with SinG: $76.8\,\mu g\,m^{-3}$). The $NO_2$ concentrations are overestimated by SinG during several time periods. Since the $NO_2/NO_x$ concentration ratio in the street with MUNICH and SinG are very similar (0.75 and 0.78 respectively), we can think that the overestimation in $NO_2$ concentrations results of the "same" error compensation than MUNICH but with higher $NO_x$ concentrations.

A sensitivity test was conducted for further investigation on the $NO_x$ underestimation with a different configuration settings and input data set (SinG-s in Table 1). As the urban background concentrations of $NO_2$ and $NO_x$ appear simulated without any strong bias with SinG (see Table B3), the uncertainties at the street level are supposed mainly related to the evaluation of the vertical transfer coefficient at roof top and to the traffic emissions data. The same modifications concerning the emissions rates, the vertical turbulent coefficient and the $NO_2/NO_x$ ratio in the emissions from traffic applied to MUNICH-s are considered

for SinG-s. Additionnaly a 33% reduction of the $O_3$ boundary conditions is applied to reduce the $NO_2/NO_x$ fraction in the simulated concentrations. The reduction of the $O_3$ boundary conditions is a pragmatic (and efficient) approach to reduce the bias in $O_3$ simulated background concentrations (see Appendix B).

The $NO_x$ concentrations of the second SinG simulation remain underestimated, however the statistical indicators are clearly improved (see Table 1). The parameters investigated deserve a more comprehensive sensitivity analysis that could be performed using a more extended observation database.

## 5.4 Analysis of SinG computational burdens

Additional simulations were conducted to estimate the increase in computational time using SinG compared to POLAIR3D. For the current case study the increase in computational burden remains limited. This is clearly due to the relatively limited fraction of the simulated domain concerned by the street-network model. The time increase using SinG is partly due to the number of iterations used to achieve steady state in MUNICH. The number of iterations depends on the set error criterion, which differs among the simulations listed as SinG-1 to SinG-5 (see Table 2). Steady state is assumed to be achieved when the errors satisfy the error criterion. This error criterion can be prescribed either in absolute terms (0.01 or 1 $\mu g\,m^{-3}$) or in relative terms (1 or 10%), with respect to the concentrations at the previous time step for all street segments of the urban canopy.

We examined the influence of the error criteria on the computational time and model results. Five additional simulations using SinG are thus compared to the one presented before using POLAIR3D as reference for the computational time. The increases of the computational time vary from 2% (SinG-5) when no error criterion is imposed (i.e., a single calculation step is conducted, for comparison it takes about 20 interations to achieve steady state in SinG-1) to 5% (SinG-3) when a 1% error criterion is imposed. Model discrepencies are estimated by comparison with the observed $NO_x$ street-canyon concentrations. Model results are not strongly influenced by changing the error limit.

The influence of the chemical kinetic mechanism on the computational time and model performance were also assessed (SinG-5 vs SinG-6). The increase of the computational time is halved when the Leighton photostationary state is used instead of CB05. Model performance is not degraded with the Leighton mechanism compared to CB05. Therefore, an operational version of SinG should use the Leighton mechanism within the urban canopy with either the SinG-2, SinG-4 or SinG-6 error criteria, depending of the accuracy desired.

## 6 Conclusions and implications

A new multi-scale model, Street-in-Grid (SinG), which combines a street-network model, Model of Urban Network of Intersecting Canyons and Highways (MUNICH), and a chemical-transport model, POLAIR3D, was developed to represent jointly the urban background and the local street-level pollution. These models were used to simulate $NO_2$ and $NO_x$ air concentrations for a Paris suburb. The simulation results were compared to background and street air concentrations measurements.

Simulation results using the street-network model MUNICH indicate that the temporal evolution of $NO_2$ and $NO_x$ concentrations in the Boulevard Alsace-Lorraine are well reproduced but $NO_2$ and $NO_x$ concentrations are underestimated. For this

**Table 2.** Comparison of the computational times and model performance for the simulated concentrations of $NO_x$ using SinG and POLAIR3D for the period from March 31 to April 6, 2014. Statistical indicators are calculated by the comparison of simulated hourly concentrations to the $NO_x$ concentrations measured at the air monitoring stations operated on the sidewalks of Boulevard Alsace-Lorraine.

| | POLAIR3D | SinG-1 | SinG-2 | SinG-3 | SinG-4 | SinG-5 | SinG-6 |
|---|---|---|---|---|---|---|---|
| Error limit[‡] | - | $\|\Delta C\| < 0.01$ $\mu g\,m^{-3}$ | $\|\Delta C\| < 1$ $\mu g\,m^{-3}$ | $\|\Delta C\|/C_0 <$ 0.01 | $\|\Delta C\|/C_0 <$ 0.1 | None | None |
| Chemistry kinetic mechanism | CB05 | CB05 | CB05 | CB05 | CB05 | CB05 | Leighton |
| Normalized computational time[†] | 1.00 | 1.04 | 1.03 | 1.05 | 1.04 | 1.02 | 1.01 |
| Observation ($\mu g\,m^{-3}$) | | | | 146.1 | | | |
| Simulation ($\mu g\,m^{-3}$) | - | 128.5 | 128.5 | 128.5 | 128.5 | 130.0 | 130.0 |
| FB[*] | - | -0.13 | -0.13 | -0.13 | -0.13 | -0.11 | -0.11 |
| $\sqrt{\text{NMSE}}$[*] | - | 0.50 | 0.50 | 0.50 | 0.50 | 0.52 | 0.51 |
| MFE[*] | - | 0.40 | 0.40 | 0.40 | 0.40 | 0.41 | 0.41 |
| VG[*] | - | 1.34 | 1.34 | 1.34 | 1.34 | 1.36 | 1.35 |
| MG[*] | - | 0.88 | 0.88 | 0.88 | 0.88 | 0.90 | 0.90 |
| FAC2[*] | - | 0.83 | 0.83 | 0.83 | 0.83 | 0.82 | 0.82 |
| R[*] | - | 0.62 | 0.62 | 0.62 | 0.62 | 0.60 | 0.60 |

[‡]: $\Delta C$ = concentration at the current time step ($C_1$) - concentration at the previous time step ($C_0$).

[†]: normalized time using POLAIR3D computational time as reference.

[*]: FB (Fractional bias), NMSE (Normal mean square error), MFE (Mean fractional error), VG (Geometrical mean squared variance), MG (Mean geometrical bias), FAC2 (Fraction in a factor of 2), R (Correlation coefficient) (Chang and Hanna, 2004; Yu et al., 2006). The statistical indicators were calculated against the observations at the monitoring stations at Boulevard Alsace-Lorraine.

case study, the use of the multi-scale model leads to a large reduction in the error and bias of the simulated concentrations in the street. Providing the background concentrations modeled by POLAIR3D to MUNICH improves the simulation results for $NO_2$ concentrations. The $NO_x$ concentrations are also improved with SinG, however both MUNICH and SinG simulated $NO_x$ concentrations are largely underestimated. This underestimation could be partly explained by uncertainties in $NO_x$ emissions or an overestimation of $NO_x$ transport into the overlying atmosphere at roof top. For this latter it would be of interest to further investigate, with the support of appropriate observation data, the relative contribution of the uncertainties in the meteorological

data and of the model assumption. The impact of the horizontal resolution of meteorological data on SinG simulations also need to be studied.

For this case study using a comprehensive chemistry within the street-canyon does not influence the $NO_x$ concentrations notably. Consequently, computational costs can be reduced by using the Leighton photostationary state within the urban canopy.

However this test would need to be renewed for new applications. The photostationary assumption cannot hold in condition with high VOC emissions. Further studies are needed to extend the model to simulate primary and secondary particulate matter in an urban canopy.

The observation database build within the framework of the TrafiPollu project was focused at the street level. We have not been able to evaluate the ability of the new model to represent background concentrations in comparison to traditional Eulerian

chemical-transport model. An application of SinG to larger urban domains would allow this type of analyse and would complete the evaluation for street level concentrations.

SinG is a useful tool to simulate both the concentrations of air pollutants in complex urban canopy configurations and the background concentrations in the overlying atmosphere. Beyond the data usually needed for CTM, traffic emissions data for street segments and urban/buildings morphology data are mandatory for a SinG simulation over an urban area. The ur-

15 ban/buildings morphology data are available for many major cities in the world (for example, ESRI ArcGIS for US, EMU for UK, OpenStreetMap). The traffic emissions may be less easily available than other data.

*Code availability.* The source code of Street-in-Grid (v1.0) is available via Zenodo with the following DOI https://doi.org/10.5281/zenodo. 1025629.

*Competing interests.* The authors declare that they have no conflict of interest.

*Acknowledgements.* This work was funded by EDF R&D and EDF R&D China. The authors acknowledge our colleagues Luc Musson-Genon, CEREA/EDF R&D, and Jiesheng Min, EDF R&D China, for helpful discussions during the model development. We also thank AIRPARIF for providing the emission inventory and the measured concentration data, Laëtitia Thouron for providing the WRF meteorological outputs, and the TrafiPollu ANR project for making those data available for the model application and evaluation.

# A   Statistical indicators

**Table A1.** Definitions of the statistical indicators.

| Indicators | Definitions |
|---|---|
| Root mean square error (RMSE) | $\sqrt{\dfrac{1}{n}\sum\limits_{i=1}^{n}(c_i - o_i)^2}$ |
| Fractional bias (FB) | $\dfrac{\overline{c} - \overline{o}}{(\overline{c} + \overline{o})/2}$ |
| Mean fractional bias (MFB) and mean fractional error (MFE) | $\dfrac{1}{n}\sum\limits_{i=1}^{n}\dfrac{c_i - o_i}{(c_i + o_i)/2}$   and   $\dfrac{1}{n}\sum\limits_{i=1}^{n}\dfrac{\mid c_i - o_i \mid}{(c_i + o_i)/2}$ |
| Mean normalized bias (MNB) and mean normalized error (MNE) | $\dfrac{1}{n}\sum\limits_{i=1}^{n}\dfrac{c_i - o_i}{o_i}$   and   $\dfrac{1}{n}\sum\limits_{i=1}^{n}\dfrac{\mid c_i - o_i \mid}{o_i}$ |
| Normalized mean square error (NMSE) | $\dfrac{\sum\limits_{i=1}^{n}(c_i - o_i)^2}{\dfrac{n}{\sum\limits_{i=1}^{n}c_i o_i}}$ |
| Correlation coefficient (R) | $\dfrac{\sum\limits_{i=1}^{n}(c_i - \overline{c})(o_i - \overline{o})}{\sqrt{\sum\limits_{i=1}^{n}(c_i - \overline{c})^2}\sqrt{\sum\limits_{i=1}^{n}(o_i - \overline{o})^2}}$ |
| Geometrical mean squared variance (VG) | $\exp\left(\dfrac{\sum\limits_{i=1}^{n}\left((\ln(c_i) - \ln(o_i))^2\right)}{n}\right)$ |
| Mean geometrical bias (MG) | $\exp\left(\dfrac{\sum\limits_{i=1}^{n}\left(\ln(c_i) - \ln(o_i)\right)}{n}\right)$ |
| Fraction of modeled values within a factor of two of observations (FAC2) | $0.5 \leq c_i/o_i \leq 2$ |

$c_i$: modeled values, $o_i$: observed values, $n$: number of data.

$$\overline{o} = \frac{1}{n}\sum_{i=1}^{n}o_i \quad \text{and} \quad \overline{c} = \frac{1}{n}\sum_{i=1}^{n}c_i$$

## B  Evaluation of simulated background concentrations

Simulated hourly concentrations of $O_3$ are compared to the concentrations measured at the background air monitoring stations on domains 2 and 3. For domain 2, $O_3$ concentrations are measured at four air monitoring stations which are operated by EMEP (see Figure 4a). Table B1 presents the comparison results. The $O_3$ conconcetrations are well estimated at a station which is located in Central France. However, the model largely overestimates the $O_3$ concentrations at threee other stations. This overestimation may be due to uncertainties in long-range $O_3$ transport. For domain 3, simulated $O_3$ concentrations are compared to the concentrations measured at six urban background monitoring stations (see Figure 4b). The modeled $O_3$ concentrations are also overestimated (MFB: 42% $\sim$ 48%) at those stations. These overestimations of $O_3$ concentrations on domains 2 and 3 at the rural and urban background stations imply uncertainties in $O_3$ boundary conditions for domain 4.

**Table B1.** Statistical indicators of the comparison of simulated hourly concentrations of $O_3$ to the concentrations measured at the background air monitoring stations within domain 2 (see Figure 4).

| Station | Observation | Simulation | MFB[*] | MFE[*] | R[*] |
|---|---|---|---|---|---|
| | ($\mu g\,m^{-3}$) | ($\mu g\,m^{-3}$) | | | |
| Revin | 78.1 | 99.1 | 0.25 | 0.28 | 0.47 |
| Morvan | 77.0 | 97.0 | 0.26 | 0.30 | 0.25 |
| Montfranc | 92.0 | 96.6 | 0.05 | 0.13 | 0.38 |
| Verneuil | 63.7 | 92.7 | 0.43 | 0.45 | 0.42 |
| Villemomble | 55.0 | 94.6 | 0.61 | 0.61 | 0.59 |
| Champigny | 56.3 | 95.1 | 0.60 | 0.60 | 0.53 |
| Les Ulis | 62.0 | 94.7 | 0.47 | 0.48 | 0.61 |
| Logne | 58.3 | 96.5 | 0.57 | 0.58 | 0.55 |
| Cergy | 60.9 | 94.6 | 0.50 | 0.51 | 0.60 |
| Neuilly-sur-Seine | 49.6 | 92.1 | 0.68 | 0.69 | 0.64 |

[*]: Mean fractional bias (MFB), mean fractional error (MFE) and correlation coefficient (R)

**Table B2.** Statistical indicators of the comparison of simulated hourly concentrations of $O_3$ to the concentrations measured at the urban background air monitoring stations within domain 3 (see Figure 4).

| Station | Observation | Simulation | MFB[*] | MFE[*] | R[*] |
|---|---|---|---|---|---|
| | ($\mu g\,m^{-3}$) | ($\mu g\,m^{-3}$) | | | |
| Villemomble | 55.0 | 82.6 | 0.47 | 0.50 | 0.69 |
| Champigny | 56.3 | 83.6 | 0.47 | 0.50 | 0.65 |
| Les Ulis | 62.0 | 91.0 | 0.42 | 0.44 | 0.64 |
| Logne | 58.3 | 87.2 | 0.48 | 0.50 | 0.66 |
| Cergy | 60.9 | 93.9 | 0.48 | 0.50 | 0.63 |
| Neuilly-sur-Seine | 49.6 | 75.2 | 0.44 | 0.51 | 0.7 |

[*]: Mean fractional bias (MFB), mean fractional error (MFE) and correlation coefficient (R)

**Table B3.** Statistical indicators of the comparison of simulated hourly concentrations of $NO_2$, $NO_x$ and $O_3$ in the SinG simulation to the concentrations measured at the urban background air monitoring stations of Villemomble and Champigny. The "$O_3$ (SinG-s)" correspond to the ozone concentrations from the simulation SinG-s using the adjusted input data including "corrected" $O_3$ boundary conditions. MFB and MFE in the $O_3$ concentration of the SinG simulation are strongly reduced using the corrected boundary conditions. However, the correlation coefficients does not change between the SinG and SinG-s simulations because the $O_3$ concentrations in the two simulations show very similar temporal evolutions.

| | Villemomble | | | | Champigny | | | |
|---|---|---|---|---|---|---|---|---|
| | $NO_x$ | $NO_2$ | $O_3$ | $O_3$ (SinG-s) | $NO_x$ | $NO_2$ | $O_3$ | $O_3$ (SinG-s) |
| Observation ($\mu g\,m^{-3}$) | 34.0 | 26.2 | 55.5 | | 36.1 | 27.7 | 56.3 | |
| Simulation ($\mu g\,m^{-3}$) | 38.9 | 30.9 | 79.4 | 51.3 | 35.3 | 28.6 | 79.6 | 56.3 |
| MFB[*] | 0.16 | 0.14 | 0.40 | -0.04 | 0.01 | -0.01 | 0.41 | -0.03 |
| MFE[*] | 0.43 | 0.39 | 0.48 | 0.4 | 0.42 | 0.39 | 0.48 | 0.39 |
| R[*] | 0.65 | 0.69 | 0.67 | 0.67 | 0.59 | 0.65 | 0.65 | 0.66 |

[*]: Mean fractional bias (MFB), mean fractional error (MFE) and correlation coefficient (R)

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
