# Peer review of "Multi-scale modeling of urban air pollution: development and application of a Street-in-Grid model (v1.0) by coupling MUNICH (v1.0) and POLAIR3D (v1.8.1)"

_Geoscientific Model Development, 2017_

## Short Comment (SC1) · 4 Oct 2017

Dear authors,

In my role as Executive editor of GMD, I would like to bring to your attention our Editorial version 1.1:

http://www.geosci-model-dev.net/8/3487/2015/gmd-8-3487-2015.html

This highlights some requirements of papers published in GMD, which is also available on the GMD website in the 'Manuscript Types' section:

http://www.geoscientific-model-development.net/submission/manuscript_types.html

In particular, please note that for your paper, the following requirements have not been met in the Discussions paper:

- "The main paper must give the model name and version number (or other unique identifier) in the title."

- "All papers must include a section, at the end of the paper, entitled 'Code availability'. Here, either instructions for obtaining the code, or the reasons why the code is not available should be clearly stated. It is preferred for the code to be uploaded as a supplement or to be made available at a data repository with an associated DOI (digital object identifier) for the exact model version described in the paper. Alternatively, for established models, there may be an existing means of accessing the code through a particular system. In this case, there must exist a means of permanently accessing the precise model version described in the paper. In some cases, authors may prefer to put models on their own website, or to act as a point of contact for obtaining the code. Given the impermanence of websites and email addresses, this is not encouraged, and authors should consider improving the availability with a more permanent arrangement. After the paper is accepted the model archive should be updated to include a link to the GMD paper."

Therefore please provide the version numbers of MUNICH and Polarir3D in the title of the manuscript. Additionally, please note that the exact code version, your article refers to, should be available, presumably via a permanent archive providing a DOI (e.g. Zenodo).

Yours,

Astrid Kerkweg

---

## Referee Comment (RC1) · Anonymous Referee #1 · 9 Oct 2017

General Comments

This is a paper to describe the development of a new Street-in-Grid (SinG) model and its initial application over a Paris suburb. Because of their inherent limitations, most current regional air quality models cannot accurately simulate the pollutant concentrations at the urban street levels. On the other hand, very few urban street network models aiming at improving urban streel-level predictions of pollutant concentrations have been developed thus far. This work fills in this critical gap through developing an urban street network model (MUNICH) and bridging it with a regional air quality model (Polair3D), it thus represents a significant scientific contribution in urban air quality modeling. The

development of SinG is technical sound and represents the state-of-the-science. SinG should be applicable to other cities in the world and can improve not only the air quality predictions at urban scale but also the accuracy of human exposure and associated health effects. The paper is well written and organized. The assumptions used in the development of MUNICH were clearly stated. For its application over a Paris suburb, SinG showed enhanced capability in representing urban street-level concentrations of major pollutants such as NO2 and O3. I would recommend acceptance of this paper for publication on GMD with minor revisions as suggested below and in the specific comments.

It would be useful to discuss uncertainties (or inaccuracies in some input data) associated with the model formulations/assumptions, input data, and the boundary conditions estimated based on limited measurements that may contribute to the underpredictions in NO2 concentrations by MUNICH and Polair3D and in NOx concentrations by MUNICH, SinG, and Polair3D. In some cases, sensitivity simulations can help pin-point the causes and estimate the relative contributions of such uncertainties to the model bias (e.g., in the application of MUNICH to a Paris suburb.

Specific Comments

1. Page 4, Section 2.1.1 described two methods to calculate the turbulent vertical mass transfer coefficients, which one is used in SinG?

2. Page 6, there are large differences in the average wind speed calculated by SIRANE and MUNICH, which one is more accurate? Have they been evaluated with observations?

3. Page 7, line 23, change "photolyses" to "photolytic reactions"

4. Page 7, line 25, Leighton photostationary state may not hold in urban air when VOCs emissions are high (e.g., morning time), this needs to be pointed out.

5. Page 8, at the end of section 2, it would be useful to briefly summarize the main

differences between SIRANE and MUNICH, in particular the strength of MUNICH over SIRANE. Also, has MUNICH been evaluated against a CFD model?

6. Page 8, line 29, is 10-min sufficiently short to represent the interactions between urban street emissions and background. Under what cases, should a shorter or longer time should be used?

7. Page 9, lines 7-16, more details on the dynamic traffic emission model used should be provided. For example, what are the species emitted from the traffic? Why was only NOx emission considered in this work? What are the uncertainties associated with calculated traffic emissions? What are the unique aspects of the dynamic traffic emission model used, comparing to static traffic emission model? Can SinG use both types of traffic emission models?

8. Page 9, lines 13-14, "Surface areas of intersections are not taken explicitly into account in MUNICH", what impact will this have on the predictions from MUNICH? Can surface areas of intersections be accounted for in future work?

9. Pages 10 and 15, sections 4.4 and 4.5, which version of WRF was used? "Satisfactory results" sounds too vague. A brief summary of the meteorological performance with some quantitative measures (e.g., NMBs, FBs, correlations) should be provided. What are the meteorological variables evaluated using observations, does it include PBLH?

10. Page 12, lines 5-8, based on section 4.4, the meteorological performance is satisfactory, what specific meteorological data may still contribute to the large discrepancies in obs. vs. sim. NO2 concentrations? Is it possible to set up a sensitivity simulation to estimate the relative contributions from uncertainties associated with calculated traffic emissions? In line 5, add "uncertainties in" before "to the model formulation or the input"? Also, since measured conc. were used to set up the background conc., the uncertainties in measured conc. may contribute to the discrepancies reported here, this should be added to the list of possible reasons.

[Figure]

11. Page 12, lines 8-13 and page 14, lines 1-4. Given the importance of background conc. and a large uncertainty in the measured conc., it may be useful to set up a sensitivity simulation to estimate the relative contributions from the uncertainties in the background conc. derived from measurements (e.g., instead of using the mean of concentrations measured at two urban background stations, using the higher conc. observed at the two stations to set up the background conc.). At minimal, some discussions on the uncertainties in limited measurements used to set up the background conc. should be discussed.

12. Page 13, Table 1, need to define the configurations used SinG-s comparing to those used in SinG in the footnote of this table.

13. Page 15, line 2, which version of MEGAN was used?

14. Page 16, lines 5-6, could you explain the meaning of "quasi-total O3 titration"? Also, what did you mean by "more limited O3 titration" which sounds confusing? Did you mean less O3 titration in SinG comparing to Polair3D?

15. Page 16, lines 9-15, Figure 7 showed that SinG tends to overpredict NO2 conc. during several time periods, what are the likely causes for those overpredictions? What are the main reasons that change the underpredictions in MUNICH to the overpredictions in SinG?

16. Page 16, line 28, add a reference for "The turbulent transfer coefficient is decreased by 25%."

17. Page 17, Were MNE and MNB calculated against Polair3D or observations? A footnote should be added to clarify this.

18. Page 1, lines 12-14, Page 18, lines 9-11 and 24-26, this is true for the test case here, but may not be always true for other cases where the Leighton photostationary state may not hold (e.g., with high VOCs that breaks down this photostationary state, which may happen in morning urban air). The abstract and conclusions need to be

modified to reflect this important point. Also, a test application over urban street networks where VOCs emissions are high (Leighton photostationary state may not hold) should be conducted in the future.

19. Page 18, it would be useful to briefly discuss the appropriateness and applicability of the SinG over other urban areas worldwide and the implications of the SinG to the quantifications of the impacts of urban traffic emissions on air quality, human exposure, and resulting health impacts.

20. Table B1, 'The "O3 cor." corresponds to the ozone concentrations from the second simulation using "corrected" boundary conditions." Does the second simulation refer to "SinG-s"? if so, the correction is not just the boundary conditions of O3, there are additional adjustments, as described in Section 5.3. Also, the R values remain the same between O3 and "O3 cor." Runs, a brief discussion on the reason should be added.
* * *

---

## Author Comment (AC1) · 7 Nov 2017

We appreciate Astrid Kerkweg for giving helpful comments to improve our manuscript.

1. Please provide the version numbers of MUNICH and Polarir3D in the title of the manuscript.

Our response:

The title of the manuscript has been corrected as follows:

Multi-scale modeling of urban air pollution: development and application of a Street-in-Grid model (v1.0) by coupling MUNICH (v1.0) and Polair3D (v1.8.1)

[Figure]

2. Additionally, please note that the exact code version, your article refers to, should be available, presumably via a permanent archive providing a DOI (e.g. Zenodo).

Our response:

The code is now available via Zenodo with the following DOI

https://doi.org/10.5281/zenodo.1025629

---

## Referee Comment (RC2) · Anonymous Referee #2 · 20 Nov 2017

General Comment:

This work is focused on the coupling of a urban street network model (MUNICH) and a regional air quality model (Polair3D), in order to develop a new Street-in-Grid (SinG) model. It was applied over a Paris suburb for a limited period (from 24th March to 14th July 2014), excluding essentially the winter period which present critical conditions for pollutant dispersion. Although the grid step size of 1 km adopted in this work is not appropriate for urban air quality modeling, SinG could represent an alternateive way to conduct it. The paper is well written and discussed. The hypothesis used in the development of MUNICH were clearly stated. I recommend acceptance of this paper

for publication on GMD, but only after major revisions as suggested below.

Major revisions:

1) The addition of urban street network model is important for the spatial pattern as well as for the temporal pattern. For this reason, long term average comparison between SinG outcomes and observations, including for instance winter months is also necessary.

2) Comparison between the meteorological model (WRF) and observations, as weel as between CTM (Polair3D) and measures are not clearly discussed. Measurements ntework included in domain 2 (Nothern and Central France) could help to ascribe discrepancies during final comment about SinG results.

Minor revisions:

- page 9, line 3. Could be useful to detail grid domain features

- page 9, lines 9-16. Only NOx are associated to traffic sources or other pollutants are considered?

- page 10, line 11. Which is the WRF version used in this work?

- page 10, line 13. As described in the major comments, WRF validation phase could be described through BIAS, CORR, IOA (Index Of Agreement). "Satisfactory results" have to be supported by statistical indexes - page 10, figure 4 - caption. Domain 1 and 2 are not clearly cited

- page 13, table 1. SinG-s configuration is not defined.

- page 15, line 2. Which is the MEGAN version?
* * *

---

## Referee Comment (RC3) · Anonymous Referee #3 · 22 Nov 2017

The paper describes the newly developed Street-in-Grid (SinG) model, for which the street-network model MUNICH has been coupled with the CTM Polair3d. Air quality models for urban areas are either used for urban background scales or street canyon scales, and a coupling of different models for different scales is often not consistent. The advantages of the SinG model presented in this work are a consistent treatment of physical and chemical processes at the different scales as well as emission input data and the influence of street level on urban background concentrations and vice versa. The paper is well written and well structured, and I recommend publication in Geoscientific Model Development with minor revisions.

[Figure]

I would ask the authors to revise the following main points (see specific comments below):

- More detailed justification and motivation for choices of the model components and the model configuration

- More details on the model evaluation, including the meteorological data used as model input

- Please include an overview table summarizing all model simulations described in this study

Specific comments:

Abstract: specify explicitly that simulations were done both with SinG and with MUNICH as stand-alone model

Section 2.2.1: I suggest moving the derivation of the alternative equation to the Annex. Instead, please motivate your choice of equation and explain why you have chosen it over the alternative. What would the differences imply for the model results? Would they have consequences for simulated concentrations?

Section 2.1.2: Please motivate the choice in wind profile descriptions. Why is the 'MUNICH' version more realistic than the 'SIRENE' version?

Page 7, line 25: "[. . .] are compared below." Please indicate where they are specified. Section 2.4: How does the dry deposition approach for the urban canopy differ from dry deposition outside urban areas, or other approaches used? Why is this one chosen?

Section 3: Please mention here that the computational cost will also be evaluated

Section 4.1: Why was this location chosen? How much time was used as spin-up of the model?

Section 4.2: Please include a more thorough description of Figure 5. Where are the main differences?

Section 4.4: Please provide more detail on the model performance of the WRF simulations. What impact would the biases in meteorological variables have on the results? Is the setup used for WRF described somewhere? I would strongly recommend a more thorough evaluation of the WRF model results if not done within a different publication yet.

Section 4.5: Please also describe the roadside measurements. Where are they located? It would be good if their location was indicated in Figure 4 or in an additional figure.

Section 4.6: How do the modeled roadside peak concentrations mentioned on page 11, lines 8 and 9, compare to observed peak concentrations? How is the diurnal cycle simulated? (daytime vs. nighttime)

Page 14, line 1: Please reformulate; in my opinion it is not possible to "replace" the measurements by simulated concentration. Rather, "base the calculations on simulated urban background concentrations" or something along those lines.

Page 15, line 12: I would suggest only using the term "significant" if you have actually done a test for statistical significance. Otherwise it should be replaced with a different formulation (e.g. considerable). This also applies to later instances in the manuscript.

Page 16, line 19: Which numbers are you comparing here in brackets? Please be more specific.

Page 16, line 23: Please specify the settings (e.g. in the table mentioned above)

Page 18, line 9: Please provide more detail on the differences in model performance. The results could for example be included in Table 1.

Figure 5: please increase the line width and size of the legend

[Figure]

Figure 6: please increase the size of the legend

[Figure]

---

## Author Response (AR1)

**Authors' reply to comments of anonymous referees on the manuscript "Multi-scale modeling of urban air pollution: development and application of a Street-in-Grid model by coupling MUNICH and Polair3D"**

Youngseob Kim[1], You Wu[2], Christian Seigneur[1], and Yelva Roustan[1]

[1]CEREA, Joint Laboratory École des Ponts ParisTech / EDF R&D, Université Paris-Est, 77455 Champs-sur-Marne, France
[2]EDF R&D China, 100005 Beijing, China

*Correspondence to:* Youngseob Kim (youngseob.kim@enpc.fr)

We appreciate the reviewers for reading the manuscript attentively and giving helpful comments to improve our manuscript.

**1 Reply to anonymous referee #1's comments**

**General comments**

This is a paper to describe the development of a new Street-in-Grid (SinG) model and its initial application over a Paris suburb. Because of their inherent limitations, most current regional air quality models cannot accurately simulate the pollutant concentrations at the urban street levels. On the other hand, very few urban street network models aiming at improving urban street-level predictions of pollutant concentrations have been developed thus far. This work fills in this critical gap through developing an urban street network model (MUNICH) and bridging it with a regional air quality model (Polair3D), it thus represents a significant scientific contribution in urban air quality modeling. The development of SinG is technical sound and represents the state-of-the-science. SinG should be applicable to other cities in the world and can improve not only the air quality predictions at urban scale but also the accuracy of human exposure and associated health effects. The paper is well written and organized. The assumptions used in the development of MUNICH were clearly stated. For its application over a Paris suburb, SinG showed enhanced capability in representing urban street-level concentrations of major pollutants such as $NO_2$ and $O_3$. I would recommend acceptance of this paper for publication on GMD with minor revisions as suggested below and in the specific comments.

It would be useful to discuss uncertainties (or inaccuracies in some input data) associated with the model formulations/assumptions, input data, and the boundary conditions estimated based on limited measurements that may contribute to the underpredictions in $NO_2$ concentrations by MUNICH and Polair3D and in $NO_x$ concentrations by MUNICH, SinG, and Polair3D. In some cases, sensitivity simulations can help pin-point the causes and estimate the relative contributions of such uncertainties to the model bias (e.g., in the application of MUNICH to a Paris suburb).

**Our response**:

We agree with the reviewer concerning the interest of sensitivity simulations. This was the aim of the discussion relative to the simulation "SinG-s" introduced in Section 5.3. With this simulation we try to emphasize the main sources of uncertainties we have identified for SinG. It is true that several of them are also relevant for MUNICH. New sensitivity simulations performed with MUNICH are then presented to illustrate this point. However through the comparison between MUNICH and SinG we show the nearest available urban background observation sites are not really representative of the air mass concentration above Boulevard Alsace Lorraine. This issue remains one of the main sources of uncertainties for MUNICH.

As the urban background concentrations of $NO_x$ appear simulated without any strong bias with SinG (statistical indicators of comparison to observation are now provided in Table B3), the uncertainties at the street level are supposed mainly related to the evaluation of the vertical transfer at roof top and to the traffic emissions data. The vertical transfer at roof top is controlled by the concentration gradient and the turbulent vertical transfer coefficient. We show that an increase of $NO_x$ emissions and a decrease of the turbulent transfer coefficient improve the simulated $NO_x$ concentrations at the street level. This modification of the model parameter also improves the concentrations simulated with MUNICH (sensitivity results for MUNICH are now included in Table 1 in the revised manuscript). The sensitivity of the model results to the turbulent transfer coefficient implies that the choice between the formulation by Salizzoni et al. (2009) and the one proposed in Schulte et al. (2015) can have an impact for streets with an aspect ratio far from 1. More studies need to be conducted for these conditions of higher aspect ratio (e.g., Paris center).

To discuss the $NO_2$ concentration simulated with SinG, we have to consider that the model in the reference configuration overestimates the $O_3$ background concentration. This overestimation of $O_3$ concentrations contributes to the overestimation of the $NO_2$ / $NO_x$ ratio. Another possible reason is the use of a too high $NO_2$ / $NO_x$ ratio for emissions. We show with the sensitivity simulation (SinG-s) that unbiased boundary conditions for $O_3$ concentrations largely improve the simulated $O_3$ background concentrations (statistical indicators of comparison to observation are now provided in Table B3). The simultaneous use of unbiased boundary conditions for $O_3$ concentrations and of a lower $NO_2$ / $NO_x$ emission ratio lead to improve the concentration ratio of $NO_2$ / $NO_x$ at the street level.

We have reorganized the discussion in Sections 4.6 and 5.3 to be more explicit concerning the main sources of uncertainties identified in this work.

**Specific comments**:

1. Page 4, Section 2.1.1 described two methods to calculate the turbulent vertical mass transfer coefficients, which one is used in SinG?

   **Our response**:

   The method proposed by Schulte et al. (2015) is used in this study. The following text has been added in the revised manuscript:

   "Here, we use the parametrization proposed by Schulte et al. (2015) for the vertical flux at roof and the exponential wind vertical profile proposed by Lemonsu et al. (2004) for the mean wind speed within the street-canyon."

2. Page 6, there are large differences in the average wind speed calculated by SIRANE and MUNICH, which one is more accurate? Have they been evaluated with observations?

    **Our response**:

    Unfortunately there is no observation data available within the framework of the TrafiPollu project to compare the results of the two methods. However, due to the relatively low aspect ratio of the street considered in this study ($\sim$ 1/3), we do not expect to have a strong sensitivity to the choice of the formulation for the average wind speed. This point could become more crucial for streets with higher aspect ratio and should be considered for future applications.

    This point is now mentioned in the text of the revised manuscript.

    "Wind speed observations were not available to compare the results of the two methods. However, due to the relatively low aspect ratio of the street considered in this study ($a_r \sim$ 1/3 for Boulevard Alsace-Lorraine), we do not expect to have a strong sensitivity to the choice of the formulation for the average wind speed. This point could become more crucial for streets with higher aspect ratio and should be considered for future applications."

3. Page 7, line 23, change "photolyses" to "photolytic reactions"

    **Our response**:

    The text has been corrected following the reviewer's comment.

4. Page 7, line 25, Leighton photostationary state may not hold in urban air when VOCs emissions are high (e.g., morning time), this needs to be pointed out.

    **Our response**:

    The following text has been added in the revised manuscript:

    "However the Leighton photostationary state may not hold even in urban environment when VOC emissions are high (Trebs et al., 2012; Matsumoto et al., 2006)."

5. Page 8, at the end of section 2, it would be useful to briefly summarize the main differences between SIRANE and MUNICH, in particular the strength of MUNICH over SIRANE. Also, has MUNICH been evaluated against a CFD model?

    **Our response**:

    A new section has been added in the revised manuscript to summarize MUNICH main characteristics and the differences with SIRANE. The concept of the street-network model MUNICH is close to the one used in SIRANE to represent concentration at the street level. We have introduced different parametrization for the vertical turbulent flux and the average wind speed. It is however not possible to definitively advocate a specific choice for these parametrization with the set of observations available within the framework of the TrafiPollu project (http://www.agence-nationale-recherche.fr/?Project=ANR-12-VBDU-0002). MUNICH is designed as a stand-alone street-network model and does not aim to

represent concentrations over the urban canopy not as SIRANE. The main strength of MUNICH over SIRANE relies on the possibility to represent a complex chemistry in the street. It also allows the interactive connection with an Eulerian chemistry transport model.

" Summary of MUNICH characteristics

The concept of the street-network model MUNICH is close to the one used in SIRANE to represent concentration at the street level. We have introduced several parametrizations for the vertical turbulent flux and the average wind speed. It is however not possible to definitively advocate a specific choice for these parametrizations with the set of observations available within the framework of the TrafiPollu project (http://www.agence-nationale-recherche.fr/?Project= ANR-12-VBDU-0002). MUNICH is then kept modular, the model can rely on the different parametrizations following user choices. MUNICH is designed as a stand-alone street-network model and does not aim to represent concentrations over the urban canopy not as SIRANE. Beyond its modularity the main strength of MUNICH over SIRANE relies on the possibility to represent a complex chemistry in the street. It also allows the interactive connection with an Eulerian chemistry transport model."

A comparison of SinG against a CFD model (Code_Saturne) is presented in the PhD thesis of Laëtitia Thouron (http://cerea.enpc.fr/fich/theses/theses_soutenues_2017/Thouron2017.pdf). Both SinG and Code_Saturne are in good agreement with the observations for averaged concentrations. SinG shows some weaknessess in reproducing the levels of concentrations when the wind speed is low (less than $1 \mathrm{~m~s}^{-1}$). An article for this comparison between SinG and Code_Saturne has been submitted to Environmental Modeling and Assessment.

6. Page 8, line 29, is 10 min sufficiently short to represent the interactions between urban street emissions and background. Under what cases, should a shorter or longer time should be used?

   **Our response**:

   In the current version, MUNICH, as SIRANE, is a stationary model. Without any change in input data, simulation results do not change with different time step. A 10 min time step for the Eulerian model appears short enough to describe the hourly evolution of the background concentrations. The other input data (emissions and meteorological fields) are available at a hourly frequency. It appears then not useful to increase the temporal resolution for the present application. It would be relevant to use finer temporal resolution with finer input data (for instance if the traffic emissions were evaluated with a shorter time step). However the time step should not be too short since the intrinsic average representation of the street-network model could become not relevant. As far as hourly emission data are available and necessary to describe the temporal evolution of hourly concentrations, we believe that a larger time step should not be considered as far as possible even in case of steady meteorological conditions.

7. Page 9, lines 7-16, more details on the dynamic traffic emission model used should be provided. For example, what are the species emitted from the traffic? Why was only $NO_x$ emission considered in this work? What are the uncertainties

associated with calculated traffic emissions? What are the unique aspects of the dynamic traffic emission model used, comparing to static traffic emission model? Can SinG use both types of traffic emission models?

**Our response**:

Traffic emissions are calculated for the species $NO_x$, CO and VOC using the dynamic traffic model and the COPERT4 emission factor. The text has been modified as follows to precise this point:

" Simulations for gas-phase species including $NO_x$, CO, VOC emissions were conducted during the period from March 24 to June 14, 2014. "

We mention the traffic model and the emission factors used in the framework of the TrafiPollu project to generate the traffic emission inventory. But we have not performed this work. For this reason we do not aim to have a comprehensive discussion on the difference between dynamic and static traffic model. We have reformulate the sentence to clarify this and add some information in the revised manuscript:

" The traffic emission inventory used for the simulation domain was built for the TrafiPollu project. This emission inventory rely on the use of the dynamic traffic model Symuvia (Leclercq et al., 2007) and the COPERT 4 emission factors (http://emisia.com/products/copert-4/versions). The dynamic traffic model Symuvia calculates the vehicle trajectories, the number of vehicles and the averaged speed on a given time period for each street segment of the simulated street network. Dynamic traffic models represent vehicle flow at smaller spatio-temporal scale than static traffic models and potentially allow an explicit representation of traffic congestion. A discussion on the differences between dynamic and static traffic models in link with water and air quality studies can be found in Shorshani et al. (2015). However for the current work the Symuvia outputs were averaged and combined with COPERT 4 emission factors to generate hourly emission rates for each street segment. The emission rates depend on the averaged vehicle speed and composition of the vehicle fleet. This latter was determined through video monitoring (André et al., 2017). It is however important to notice that the vehicle fleet composition appears to be a sensitive input data (Carteret et al., 2014; Chen et al., 2017). "

As an input of SinG (MUNICH) traffic emissions can be provided by any type of traffic emission data source as far as an information is provided for each street segment. There is no explicit link between a given emission modelling chain or database and the dispersion model presented here.

Concerning the uncertainties associated with calculated traffic emissions we add the following text:

" Since the traffic model is calibrated with flow observation and the vehicle fleet composition determined through video monitoring, the remaining uncertainties in the emission data lie in the use of only two typical days to represent the whole period and in COPERT 4 emission factors. "

8. Page 9, lines 13-14, "Surface areas of intersections are not taken explicitly into account in MUNICH", what impact will this have on the predictions from MUNICH? Can surface areas of intersections be accounted for in future work?

**Our response**:

We add the following text to emphasize this point:

"The geometry of the intersection can influence the mass exchange (Salem et al., 2015). In particular, when intersections are large, vertical mixing with the overlying atmosphere becomes more important. As this phenomenon is not taken into account in the current version of the model it leads to underestimate the exchanges through such open space in the street network. There is a need here to extend the modeling framework to better represent this type of urban space."

To our knowledge no specific parametrization is currently proposed to represent intersections. Such a parametrization could probably be developed from field or physical model experiments and CFD modeling studies as it has been done to develop SIRANE for instance. However due to the complexity of the flow and the diversity of situations it would not be straightforward.

9. Pages 10 and 15, sections 4.4 and 4.5, which version of WRF was used? "Satisfactory results" sounds too vague. A brief summary of the meteorological performance with some quantitative measures (e.g., NMBs, FBs, correlations) should be provided. What are the meteorological variables evaluated using observations, does it include PBLH?

    **Our response**:

    WRF version 3.6.1 is used. The version is cited in the revised manuscript. The modeled PBLH is not compared because observation data are not available for the monitoring stations.

    The following text has been added in the revised manuscript.

    " The root-mean square error (RMSE), the fractional bias (FB), and the correlation coefficient (R) are the statistical indicators used in Thouron et al. (2017) to evaluate the meteorological fields. The WRF simulation slightly overestimates the temperature (RMSE: $0.2 \sim 1.1$ °C, FB: $0.02 \sim 0.07$ and R: 0.9) and overestimate the wind speed (RMSE: $0.8 \sim 1.1 \, \mathrm{m\,s^{-1}}$, FB: $0.2 \sim 0.3$ and R: $0.6 \sim 0.7$). The modeled wind direction is biased by an angular differences of about 15°. An important error in the precipitation modeling is obtained (RMSE: $0.04 \, \mathrm{mm\,h^{-1}}$, FB: -0.6, R: 0.1) but this model error has not a strong impact on the concentration of the poorly soluble species simulated. "

10. Page 12, lines 5-8, based on section 4.4, the meteorological performance is satisfactory, what specific meteorological data may still contribute to the large discrepancies in obs. vs. sim. NO2 concentrations? Is it possible to set up a sensitivity simulation to estimate the relative contributions from uncertainties associated with calculated traffic emissions? In line 5, add "uncertainties in" before "to the model formulation or the input"? Also, since measured conc. were used to set up the background conc., the uncertainties in measured conc. may contribute to the discrepancies reported here, this should be added to the list of possible reasons.

    **Our response**:

    The $NO_x$ transport into the overlying atmosphere at roof top appears as an important source of uncertainties. The standard deviation of the vertical wind velocity ($\sigma_w$) at roof level is necessary to compute the vertical transport. $\sigma_w$ depends on

the friction velocity, the Monin-Obukhov length and PBLH which contribute to the global uncertainty. However as no observation are available for these fields, it is difficult to propose more than a global sensitivity test through the turbulent dispersion coefficient.

As previously mentioned a new simulation (MUNICH-s) has been performed and the text has been modified and reorganised to follow the reviewer's comment. We have reorganised the discussion in Sections 4.6 and 5.3 to be more explicit concerning the main sources of uncertainties identified in this work.

11. Page 12, lines 8-13 and page 14, lines 1-4. Given the importance of background conc. and a large uncertainty in the measured conc., it may be useful to set up a sensitivity simulation to estimate the relative contributions from the uncertainties in the background conc. derived from measurements (e.g., instead of using the mean of concentrations measured at two urban background stations, using the higher conc. observed at the two stations to set up the background conc.). At minimal, some discussions on the uncertainties in limited measurements used to set up the background conc. should be discussed.

    **Our response**:

    Following the reviewer's comment two additional simulations were conducted and the text below has been added in the section :

    " Two additional simulations were conducted to assess the relative contributions from the uncertainties in the background concentrations derived from measurements. For $NO_2$, $NO_x$ and $O_3$ the standard deviations over the simulated period of the differences between the measured concentrations at the two monitoring stations are calculated ($\sigma_{NO}$: $8.1\,\mu g\,m^{-3}$, $\sigma_{NO_2}$: $6.5\,\mu g\,m^{-3}$ and $\sigma_{O_3}$: $5.1\,\mu g\,m^{-3}$). The first simulation was run with $O_3$ concentrations increased by $\sigma_{O_3}$ and NO and $NO_2$ concentrations lowered by $\sigma_{NO}$ and $\sigma_{NO_2}$ respectively. In the second simulation reduced $O_3$ concentration and increased NO and $NO_2$ concentrations are used. Differences between the averaged $NO_2$ concentrations for these simulations and the reference simulation are up to 30%."

12. Page 13, Table 1, need to define the configurations used SinG-s comparing to those used in SinG in the footnote of this table.

    **Our response**:

    The footnote of the table for the SinG-s configurations is added in the revised manuscript as follows:

    " ***: For the simulation "SinG-s" a 25% decrease of the turbulent transfer coefficient, a 33% reduction of the $O_3$ boundary conditions, a one-third increase of $NO_x$ emissions from traffic and a reduction from 20% to 9% of the $NO_2/NO_x$ emissions ratio (in mass of $NO_2$ equivalent) are applied. "

13. Page 15, line 2, which version of MEGAN was used?

    **Our response**:

    The version of MEGAN is 2.04 and it is added in the revised manuscript.

14. Page 16, lines 5-6, could you explain the meaning of "quasi-total O3 titration"? Also, what did you mean by "more limited O3 titration" which sounds confusing? Did you mean less O3 titration in SinG comparing to Polair3D?

   **Our response**:

   The text has been corrected as follows:

   " It is due to less $O_3$ titration in SinG than in Polair3D. In SinG, vertical dispersion of $NO_x$ is constrained by the urban canopy. Therefore, $O_3$ titration is less in SinG in comparison to Polair3D due to lower NO concentrations above the urban canopy. "

15. Page 16, lines 9-15, Figure 7 showed that SinG tends to overpredict NO2 conc. during several time periods, what are the likely causes for those overpredictions? What are the main reasons that change the underpredictions in MUNICH to the overpredictions in SinG?

   **Our response**:

   Since the $NO_2/NO_x$ concentration ratio in the street with MUNICH and SinG are very similar (0.75 and 0.78 respectively), we can think that the overestimation in $NO_2$ concentrations results of the "same" error compensation than MUNICH but with higher $NO_x$ concentrations.

   The following text has been added in the revised manuscript.

   " The $NO_2$ concentrations are overestimated by SinG during several time periods. Since the $NO_2/NO_x$ concentration ratio in the street with MUNICH and SinG are very similar (0.75 and 0.78 respectively) we can think that the overestimation in $NO_2$ concentrations results of the "same" error compensation than MUNICH but with higher $NO_x$ concentrations. "

16. Page 16, line 28, add a reference for "The turbulent transfer coefficient is decreased by 25%."

   **Our response**:

   Due to the reorganisation of the discussion this point appear now in Section 4.6. The following text was added to explain the choice of a 25% decrease.

   " The magnitude of the turbulent transfer coefficient decrease is somewhat arbitrary. It is however chosen consistent with the difference between the two parametrizations considered for the vertical turbulent transfer (Figure 1) for the aspect ratio of Boulevard Alsace-Lorraine. "

17. Page 17, Were MNE and MNB calculated against Polair3D or observations? A footnote should be added to clarify this.

   **Our response**:

   The statistical indicators for the $NO_x$ concentrations were calculated between the SinG simulation results and the observations at the monitoring stations at Boulevard Alsace-Lorraine. The footnote has been corrected as follows:

   " *: FB (Fractional bias), NMSE (Normal mean square error), MFE (Mean fractional error), VG (Geometrical mean squared variance), MG (Mean geometrical bias), FAC2 (Fraction in a factor of 2), R (Correlation coefficient) (Chang

and Hanna, 2004; Yu et al., 2006). The statistical indicators were calculated against the observations at the monitoring stations at Boulevard Alsace-Lorraine. "

18. Page 1, lines 12-14, Page 18, lines 9-11 and 24-26, this is true for the test case here, but may not be always true for other cases where the Leighton photostationary state may not hold (e.g., with high VOCs that breaks down this photostationary state, which may happen in morning urban air). The abstract and conclusions need to be modified to reflect this important point. Also, a test application over urban street networks where VOCs emissions are high (Leighton photostationary state may not hold) should be conducted in the future.

**Our response**:

The manuscript has been corrected following the reviewer's comment.

For the abstract, " For the case study considered, the model performance for $NO_x$ concentrations is not sensitive to using a complex chemistry model in MUNICH and the Leighton $NO/NO_2/O_3$ set of reactions is sufficient. "

And the text,

" Using a comprehensive chemistry within the street-canyon does not influence the $NO_x$ concentrations notably in this study. Consequently, computational costs can be reduced significantly by using the Leighton photostationary state within the urban canopy. However an another test need to be conducted under the condition where VOC emissions are high. Further studies are needed to extend the model to simulate primary and secondary particulate matter in an urban canopy. "

19. Page 18, it would be useful to briefly discuss the appropriateness and applicability of the SinG over other urban areas worldwide and the implications of the SinG to the quantifications of the impacts of urban traffic emissions on air quality, human exposure, and resulting health impacts.

**Our response**:

The following text has been added in the revised manuscript.

" SinG is a useful tool to simulate both the concentrations of air pollutants in complex urban canopy configurations and the background concentrations in the overlying atmosphere. Beyond the data usually needed for CTM, traffic emissions data for street segments and urban/buildings morphology data are mandatory for a SinG simulation over an urban area. The urban/buildings morphology data are available for many major cities in the world (for example, ESRI ArcGIS for US, EMU for UK, OpenStreetMap). The traffic emissions may be less easily available than other data. "

20. Table B1, 'The "O3 cor." corresponds to the ozone concentrations from the second simulation using "corrected" boundary conditions." Does the second simulation refer to "SinG-s"? if so, the correction is not just the boundary conditions of O3, there are additional adjustments, as described in Section 5.3. Also, the R values remain the same between O3 and "O3 cor." Runs, a brief discussion on the reason should be added.

**Our response**:

The footnote in Table B1 (Table B3 in the revised manuscript) has been corrected as follows:

" Statistical indicators of the comparison of simulated hourly concentrations of $NO_2$, $NO_x$ and $O_3$ in the SinG simulation to the concentrations measured at the urban background air monitoring stations of Villemomble and Champigny. The "$O_3$ (SinG-s)" correspond to the ozone concentrations from the simulation SinG-s using the adjusted input data including "corrected" $O_3$ boundary conditions. MFB and MFE in the $O_3$ concentration of the SinG simulation are significantly reduced using the corrected boundary conditions. However, the correlation coefficients does not change between the SinG and SinG-s simulations because the $O_3$ concentrations in the two simulations show very similar temporal evolutions. "

**2  Reply to anonymous referee #2's comments**

**General comments**

This work is focused on the coupling of a urban street network model (MUNICH) and a regional air quality model (Polair3D), in order to develop a new Street-in-Grid (SinG) model. It was applied over a Paris suburb for a limited period (from 24th March to 14th July 2014), excluding essentially the winter period which present critical conditions for pollutant dispersion. Although the grid step size of 1 km adopted in this work is not appropriate for urban air quality modeling, SinG could represent an alternative way to conduct it. The paper is well written and discussed. The hypothesis used in the development of MUNICH were clearly stated. I recommend acceptance of this paper for publication on GMD, but only after major revisions as suggested below.

**Major revisions**:

1. The addition of urban street network model is important for the spatial pattern as well as for the temporal pattern. For this reason, long term average comparison between SinG outcomes and observations, including for instance winter months is also necessary.

   **Our response**:

   The first case study chosen to support the development of MUNICH and SinG correspond to a database built in the framework of the TrafiPollu project (http://www.agence-nationale-recherche.fr/?Project=ANR-12-VBDU-0002). We have used all the observations available within this framework to evaluate our model development. We fully agree that this evaluation is however not comprehensive, but we believe it is enough to prove the interest of the SinG concept. From this first evaluation our aim is of course to continue the model development and complete its evaluation over longer period. And including winter period is indeed relevant since peaks of pollutants concentrations during winter time are regularly observed over urban areas. This will be done in future project.

   We have emphasized in the text and in the conclusion that further studies are needed to evaluate some modeling aspect (e.g. behavior of the model for street segment with higher aspect ratio) or simply enlarge the possible application (e.g. include the treatment of particulate matter).

2. Comparison between the meteorological model (WRF) and observations, as well as between CTM (Polair3D) and measures are not clearly discussed. Measurements network included in domain 2 (Northern and Central France) could help to ascribe discrepancies during final comment about SinG results.

**Our response**:

A discussion for the comparison between WRF and observations has been added in the revised manuscript as follows:

" The root-mean square error (RMSE), the fractional bias (FB), and the correlation coefficient (R) are the statistical indicators used in Thouron et al. (2017) to evaluate the meteorological fields. The WRF simulation slightly overestimates the temperature (RMSE: $0.2 \sim 1.1$ °C, FB: $0.02 \sim 0.07$ and R: 0.9) and overestimate the wind speed (RMSE: $0.8 \sim 1.1\,\mathrm{m\,s^{-1}}$, FB: $0.2 \sim 0.3$ and R: $0.6 \sim 0.7$). The modeled wind direction is biased by an angular differences of about $15°$. An important error in the precipitation modeling is obtained (RMSE: $0.04\,\mathrm{mm\,h^{-1}}$, FB: -0.6, R: 0.1) but this model error has not a strong impact on the concentration of the poorly soluble species simulated. "

A discussion for the comparison between Polair3D and observations on Domain 2 has been added in the revised manuscript as follows:

" Simulated hourly concentrations of $O_3$ are compared to the concentrations measured at the background air monitoring stations on domains 2 and 3. For domain 2, $O_3$ concentrations are measured at four air monitoring stations which are

[Figure]

[Figure]

**Figure 1.** Four simulation domains are simulated from the continental scale to the urban scale. In the left panel, the largest domain 1 covers western Europe. Domain 2 covers northern/central France. the red circles show the locations of the background air monitoring stations. In the right panel, domains 3 and 4 cover the Île-de-France region and the eastern Paris suburbs. the blue box corresponds to the modeling area in suburban Paris for the MUNICH simulations. The black stars and red circles show the locations of the urban background air monitoring stations. Measured data at the stations with the black stars are used for background concentrations in the MUNICH simulations. SinG is only used for domain 4.

operated by EMEP (see Figure 1a). Table 1 presents the comparison results. The $O_3$ concentrations are well estimated at a station which is located in Central France. However, the model significantly overestimates the $O_3$ concentrations at three other stations. This overestimation may be due to uncertainties in long-range $O_3$ transport. For domain 3, simulated $O_3$ concentrations are compared to the concentrations measured at six urban background monitoring stations (see Figure 1b). The modeled $O_3$ concentrations are also overestimated (MFB: $42\% \sim 48\%$) at those stations. These overestimations of $O_3$ concentrations on domains 2 and 3 at the rural and urban background stations imply uncertainties in $O_3$ boundary conditions for domain 4."

**Minor revisions**:

3. page 9, line 3. Could be useful to detail grid domain features

   **Our response**:

   The text has been corrected in the revised manuscript as follows:

   " MUNICH was applied to simulate the concentrations of pollutants in a Paris suburb (Le Perreux-sur-Marne, 13 km east of Paris). Figure 4 displays the location of the modeling domain. The street-network within the simulation domain consists of 577 street segments and is displayed in Figure 5. "

4. page 9, lines 9-16. Only NOx are associated to traffic sources or other pollutants are considered?

   **Our response**:

**Table 1.** Statistical indicators of the comparison of simulated hourly concentrations of $O_3$ to the concentrations measured at the background air monitoring stations within domain 2 (see Figure 1).

| Station | Observation | Simulation | MFB* | MFE* | R* |
|---------|-------------|------------|------|------|-----|
| | ($\mu g\,m^{-3}$) | ($\mu g\,m^{-3}$) | | | |
| Revin | 78.1 | 99.1 | 0.25 | 0.28 | 0.47 |
| Morvan | 77.0 | 97.0 | 0.26 | 0.30 | 0.25 |
| Montfranc | 92.0 | 96.6 | 0.05 | 0.13 | 0.38 |
| Verneuil | 63.7 | 92.7 | 0.43 | 0.45 | 0.42 |
| Villemomble | 55.0 | 94.6 | 0.61 | 0.61 | 0.59 |
| Champigny | 56.3 | 95.1 | 0.60 | 0.60 | 0.53 |
| Les Ulis | 62.0 | 94.7 | 0.47 | 0.48 | 0.61 |
| Logne | 58.3 | 96.5 | 0.57 | 0.58 | 0.55 |
| Cergy | 60.9 | 94.6 | 0.50 | 0.51 | 0.60 |
| Neuilly-sur-Seine | 49.6 | 92.1 | 0.68 | 0.69 | 0.64 |

*: Mean fractional bias (MFB), mean fractional error (MFE) and correlation coefficient (R)

Traffic emissions are calculated for the species $NO_x$, CO and VOC using the dynamic traffic model and the COPERT4 emission factor.

The text has been corrected as follows:

" Simulations for gas-phase species including $NO_x$, CO, VOC emissions were conducted "

5. page 10, line 11. Which is the WRF version used in this work?

   **Our response**:

   WRF version 3.6.1 is used. The version is cited in the revised manuscript.

6. page 10, line 13. As described in the major comments, WRF validation phase could be described through BIAS, CORR, IOA (Index Of Agreement). "Satisfactory results" have to be supported by statistical indexes

   **Our response**:

   As previously mentioned a discussion on WRF validation has been added in the revised manuscript.

7. page 10, figure 4 - caption. Domain 1 and 2 are not clearly cited

   **Our response**:

   The caption of figure 4 has been corrected in the revised manuscript as follows:

   " Four simulation domains for the continental scale to the urban scale. In the left panel, the largest domain 1 covers western Europe. Domain 2 covers northern/central France. The red circles show the locations of the background air monitoring stations. In the right panel, domains 3 and 4 cover the Île-de-France region and the eastern Paris suburbs. The blue box corresponds to the modeling area in suburban Paris for the MUNICH simulations. The black stars and red circles show the locations of the urban background air monitoring stations. SinG is only used for domain 4. Measured data at the stations with the black stars are used for background concentrations in the MUNICH simulations. "

8. page 13, table 1. SinG-s configuration is not defined.

   **Our response**:

   The footnote of the table for the SinG-s configurations is added in the revised manuscript as follows:

   " ***: For the simulation "SinG-s" a 25% decrease of the turbulent transfer coefficient, a 33% reduction of the $O_3$ boundary conditions, a one-third increase of $NO_x$ emissions from traffic and a reduction from 20% to 9% of the $NO_2/NO_x$ emissions ratio (in mass of $NO_2$ equivalent) are applied. "

9. page 15, line 2. Which is the MEGAN version?

   **Our response**:

   The version of MEGAN is 2.04 and it is added in the revised manuscript.

**3 Reply to anonymous referee #3's comments**

**General comments**

The paper describes the newly developed Street-in-Grid (SinG) model, for which the street-network model MUNICH has been coupled with the CTM Polair3d. Air quality models for urban areas are either used for urban background scales or street canyon scales, and a coupling of different models for different scales is often not consistent. The advantages of the SinG model presented in this work are a consistent treatment of physical and chemical processes at the different scales as well as emission input data and the influence of street level on urban background concentrations and vice versa. The paper is well written and well structured, and I recommend publication in Geoscientific Model Development with minor revisions.

**Minor revisions**

1. Section 4.2: Please include a more thorough description of Figure 5. Where are the main differences?

   **Our response**:

   The following text has been added in the revised manuscript.

   " Figure 5 shows the $NO_x$ traffic emissions which were estimated using the dynamic traffic model with the COPERT 4 emission factors on the simulation domain in the Paris suburb. In the left panel, $NO_x$ emission rates during nighttime are presented. very low emission rates are estimated for all the streets even though those on the A86 highway are slightly higher. In the right panel, $NO_x$ emission rates during morning rush-hour increase more than $1400\,\mu g\,m^{-1}\,s^{-1}$. "

2. Section 4.4: Please provide more detail on the model performance of the WRF simulations. What impact would the biases in meteorological variables have on the results? Is the setup used for WRF described somewhere? I would strongly recommend a more thorough evaluation of the WRF model results if not done within a different publication yet.

**Our response**:

A discussion for the comparison between WRF and observations has been added in the revised manuscript as follows:

" The root-mean square error (RMSE), the fractional bias (FB), and the correlation coefficient (R) are the statistical indicators used in Thouron et al. (2017) to evaluate the meteorological fields. The WRF simulation slightly overestimates the temperature (RMSE: $0.2 \sim 1.1$ °C, FB: $0.02 \sim 0.07$ and R: 0.9) and overestimate the wind speed (RMSE: $0.8 \sim 1.1\,\mathrm{m\,s}^{-1}$, FB: $0.2 \sim 0.3$ and R: $0.6 \sim 0.7$). The modeled wind direction is biased by an angular differences of about $15°$. An important error in the precipitation modeling is obtained (RMSE: $0.04\,\mathrm{mm\,h}^{-1}$, FB: -0.6, R: 0.1) but this model error has not a strong impact on the concentration of the poorly soluble species simulated. "

3. Section 4.5: Please also describe the roadside measurements. Where are they located? It would be good if their location was indicated in Figure 4 or in an additional figure.

**Our response**:

The location of the air monitoring stations on the sidewalks is indicated in Figure 6.

4. Section 4.6: How do the modeled roadside peak concentrations mentioned on page 11, lines 8 and 9, compare to observed peak concentrations? How is the diurnal cycle simulated? (daytime vs. nighttime)

**Our response**:

The following figure and text for the diurnal cycle have been added in the revised manuscript.

[Figure]

**Figure 2.** Diurnal variation of $NO_2$ concentrations modeled with MUNICH (blue line), Polair3D (green line) and the SinG model (red line). They are compared to the measured concentrations (black line) at the stations nearby traffic on each sidewalks of Boulevard Alsace-Lorraine.

" Mean diurnal variations of $NO_2$ concentrations over this period are presented in Figure 2. Statistical indicators defined in Appendix 1 for the comparison of hourly concentrations are provided in Table 1. The $NO_2$ modeled concentrations using MUNICH generally underestimate the observations with a mean negative bias of 32%. Simulated morning and evening peaks are delayed compared to the observation. The morning peak of emissions data for the street segment of Boulevard Alsace-Lorraine corresponds in time to the peak of observed concentrations. It is also important to note that in average over the street network the morning peak of emissions data occurs one hour later than in Boulevard Alsace-Lorraine. It means that the delay in simulated concentrations is introduced by a transport process (advection in the street network or turbulent exchange with the background atmosphere). "

5. Page 14, line 1: Please reformulate; in my opinion it is not possible to "replace" the measurements by simulated concentration. Rather, "base the calculations on simulated urban background concentrations" or something along those lines.

   **Our response**:

   The text has been reformulated as follows:

   " As shown in the following the urban background concentrations can be estimated based on the concentrations simulated with an Eulerian model. "

6. Page 15, line 12: I would suggest only using the term "significant" if you have actually done a test for statistical significance. Otherwise it should be replaced with a different formulation (e.g. considerable). This also applies to later instances in the manuscript.

   **Our response**:

   The text has been reformulated following the reviewer's suggestion.

7. Page 16, line 19: Which numbers are you comparing here in brackets? Please be more specific.

   **Our response**:

   The text has been rewritten as follows:

   " (measurement: $148.5\,\mu g\,m^{-3}$ and simulation with SinG: $76.8\,\mu g\,m^{-3}$). "

8. Page 16, line 23: Please specify the settings (e.g. in the table mentioned above)

   **Our response**:

   The footnote of the table for the SinG-s configurations is added in the revised manuscript as follows:

   " ***: For the simulation "SinG-s" a 25% decrease of the turbulent transfer coefficient, a 33% reduction of the $O_3$ boundary conditions, a one-third increase of $NO_x$ emissions from traffic and a reduction from 20% to 9% of the $NO_2/NO_x$ emissions ratio (in mass of $NO_2$ equivalent) are applied. "

9. Page 18, line 9: Please provide more detail on the differences in model performance. The results could for example be included in Table 1.

   **Our response**:

   Table 1 (Table 2 in the manuscript) has been corrected in the revised manuscript.

10. Figure 5: please increase the line width and size of the legend

    **Our response**:

    The figure has been corrected following the reviewer's suggestion.

11. Figure 6: please increase the size of the legend

   **Our response**:

   The figure has been corrected following the reviewer's suggestion.

[revised manuscript text omitted]